# Towards Robustness Certification Against Universal Perturbations

Yi Zeng [*1], Zhouxing Shi [*2], Ming Jin[1], Feiyang Kang[1], Lingjuan Lyu[3],
Cho-Jui Hsieh[2] and Ruoxi Jia[1]

[1]Virginia Tech, Blacksburg, VA 24061, USA
[2]University of California, Los Angeles, CA 90095, USA
[3]Sony AI, Tokyo, 108-0075, Japan

## Abstract

In this paper, we investigate the problem of certifying neural network robustness against universal perturbations (UPs), which have been widely used in universal adversarial attacks and backdoor attacks. Existing robustness certification methods aim to provide robustness guarantees for each sample with respect to the worst-case perturbations given a neural network. However, those sample-wise bounds will be loose when considering the UP threat model as they overlook the important constraint that the perturbation should be shared across all samples. We propose a method based on a combination of linear relaxation-based perturbation analysis and Mixed Integer Linear Programming to establish the first robust certification method for UP. In addition, we develop a theoretical framework for computing error bounds on the entire population using the certification results from a randomly sampled batch. Aside from an extensive evaluation of the proposed certification, we further show how the certification facilitates efficient comparison of robustness among different models or efficacy among different universal adversarial attack defenses and enables accurate detection of backdoor target classes.

## 1 Introduction

As deep neural networks become prevalent in modern performance-critical systems such as self-driving cars and healthcare, it is critical to understand their failure modes and performance guarantees. Universal perturbations (UPs) are an important class of vulnerabilities faced by deep neural networks. Such perturbations can fool a classifier into misclassifying *any* input from a given distribution with high probability at test time. Past literature has studied two lines of techniques to create UPs: universal adversarial attacks (Moosavi-Dezfooli et al., 2017) and backdoor attacks (Gu et al., 2019; Chen et al., 2017). The former *crafts* a UP based on a trained model and does *not* rely on access to training data. The latter, by contrast, *prespecifies* a pattern as a UP and further alters the training data so that adding the pattern (often known as the trigger in backdoor attack literature) will change the output of the trained classifier into an attacker-desired target class.

Many defenses have been proposed for both universal adversarial attacks (Akhtar & Mian, 2018; Moosavi-Dezfooli et al., 2017; Shafahi et al., 2020; Benz et al., 2021; Liu et al., 2021) and backdoor attacks (Wang et al., 2019; Chen et al., 2019; Guo et al., 2019; Borgnia et al., 2020; Qiu et al., 2021). But empirical evaluation with attacks does not provide a formal guarantee on the robustness as it is infeasible for an attack algorithm to provably cover all concerned perturbations. In contrast, *robustness certification* aims to verify the output bounds of the model given a certain class of input perturbations and provably certify the robustness against all the concerned perturbations. Although several recent works (Weber et al., 2020; Xie et al., 2021) developed techniques to achieve certified robustness of a classifier against backdoor-attack-induced UPs with certain norm bound. However, these techniques apply to specific learning algorithms and require the knowledge of the training data. It remains an open question:

> How to certify the robustness of a trained model against a class of UPs in a way that is agnostic to the underlying training algorithm and data, and is general for different UPs (including both universal adversarial attacks and norm-bounded backdoor attacks)?

---

*Zhouxing Shi and Yi Zeng contributed equally. Corresponding Yi Zeng, Lingjuan Lyu or Ruoxi Jia. Work partially done during Yi Zeng's internship at Sony AI.

In this paper, we propose a framework to certify the worst-case classification accuracy on a batch of test samples against $l_\infty$-norm-bounded UPs. Our approach builds off of past works for certifying robustness against sample-wise perturbations that are independently added to each sample. For efficient verification, many recent works linearly relax nonlinear activation functions in neural networks into linear bounds and then conduct linear bound propagation to obtain the output bounds for the whole model (Wong & Kolter, 2018; Wang et al., 2018b; Dvijotham et al., 2018; Zhang et al., 2018; Singh et al., 2019b). This process is also referred to as *linear perturbation analysis* (Xu et al., 2020a). Since the worst-case model accuracy against sample-wise perturbations is a lower bound of the worst-case accuracy against UPs, these certification techniques could be applied to obtain a certificate against UPs. However, a direct application would overlook the important constraint that a UP is shared across different inputs, thereby producing overly conservative certification results.

Unlike sample-wise perturbations, UPs require theoretical reasoning to generalize certification results. This is because UPs are applied to any input from the data distribution, and our main interest lies in the expected model accuracy over the entire data distribution against UPs. However, certification procedures can only accept a batch of samples from the distribution and certify the accuracy over the samples. Therefore, it's crucial to understand the discrepancy between certified robustness computed from samples and the actual population robustness.

We summarize our contributions as follows:

- We formulate the problem of robustness certification against UPs. We then generalize linear relaxation based perturbation analysis (LiRPA) to UPs, and we further propose a Mixed Integer Linear Programming (MILP) formulation over linear bounds from LiRPA, to obtain tighter certification on the worst-case accuracy of a given model against UPs within a $\ell_\infty$-norm ball[1].
- We establish a theoretical framework for analyzing the generalizability of the certification results based on random sampled subsets to the entire population.
- We conduct extensive experiments to show that our certification method provides certified lower bounds on the worst-case robust accuracy against both universal adversarial attacks and $l_\infty$-bounded backdoor attacks, which are substantially tighter than results by directly applying existing sample-wise certification.
- We also investigate the implications of robustness certification on UPs to facilitate easy comparisons of robustness among different models or the efficacy of empirical defenses, and to achieve reliable identification of backdoor target classes.

## 2 BACKGROUND AND RELATED WORK

**Universal Adversarial Perturbation** Neural networks are vulnerable to adversarial examples (Szegedy et al., 2014), which has led to the development of universal adversarial perturbations (UAPs), a same noise can consistently deceive a target network on most images (Liu et al., 2019; 2020). Existing defenses against UAPs include fine-tuning on pre-computed UAPs (Moosavi-Dezfooli et al., 2017), post-hoc detection (Akhtar et al., 2018), universal adversarial training with online UAP generation (Mummadi et al., 2019; Shafahi et al., 2020; Benz et al., 2021). However, all existing defenses to UAPs are empirical works without efficacy guarantee to new attacks.

**Backdoor Attacks** In backdoor attacks, attackers plant a predefined UP (a.k.a. the trigger) in the victim model by manipulating the training procedure (Li et al., 2020c). Attacked models can give adversarially-desired outputs for any input patched with the trigger while still show good performance on clean inputs. Existing defenses include: poison detection via outlier detection (Gao et al., 2019; Chen et al., 2018; Tran et al., 2018; Zeng et al., 2021) which rely on the modeling of clean samples' distribution; poisoned model identification (Xu et al., 2019; Wang et al., 2020b); trojan removal via trigger synthesising (Wang et al., 2019; Chen et al., 2019; Guo et al., 2019; Zeng et al., 2022a), or preprocessing and fine-tuning; (Li et al., 2020b; Borgnia et al., 2020); robust training via differential privacy (Du et al., 2019) or redesigning the training pipeline (Levine & Feizi, 2020; Jia et al., 2020; Huang et al., 2022; Li et al., 2021). As all these defenses were empirical, existing literature has revealed those empirical defenses' limitations to zero-day attacks or adaptive attacks (Zeng et al., 2022b).

**Robustness Certification of Neural Networks** Early robustness certifications (Katz et al., 2017; Ehlers, 2017; Tjeng et al., 2017) largely relied on satisfiability modulo theory (SMT) or integer

---

[1] https://github.com/ruoxi-jia-group/Universal_Pert_Cert

linear programming (ILP) solvers are were limited to very small networks. For more efficient verification, bound propagation with convex relaxations has been proposed (Wong & Kolter, 2018; Wang et al., 2018b; Zhang et al., 2018; Weng et al., 2018; Singh et al., 2019b; Salman et al., 2019), which over-approximates nonlinear activations with convex relaxation and propagates the bounds layer by layer to finally bound the entire model. Xu et al. (2020a) proposed a bound propagation framework for general computational graphs and referred to the related methods as linear relaxation based perturbation analysis (LiRPA), as activations are relaxed by linear bounds. Bound propagation methods have also been further enhanced with techniques such as branch-and-bound (Bunel et al., 2018; 2020; Wang et al., 2018a;b; Xu et al., 2020b; Wang et al., 2021), multi-neuron relaxation and cutting planes (Singh et al., 2019a; Ferrari et al., 2021; Zhang et al., 2022a) for tighter results at a cost of efficiency. However, these works are developed for sample-wise perturbations, and they cannot directly produce tight certification against universal perturbations. Besides, there are several randomized smoothing (Cohen et al., 2019) based methods for certified robustness against backdoor attacks (Weber et al., 2020; Wang et al., 2020a; Xie et al., 2021; Zhang et al., 2022b). These are stochastic methods and are usually considered orthogonal to deterministic certification. Moreover, they require access to training data, only applicable to some specific learning algorithms (e.g., binary models or federated learning) and not general for other UPs, such as UAPs.

## 3 METHODOLOGY

### 3.1 PROBLEM FORMULATION

On a set of $n$ independent samples $\{z^{(1)}, \ldots, z^{(n)}\}$ from the data distribution $\Omega$, where $z^{(i)} = (\mathbf{x}_i, y_i)$ is the $i$-th example, $\mathbf{x}_i$ ($\mathbf{x}_i \in \mathbb{R}^d$) is the input and $y_i$ is the ground-truth label, we aim to certify the robustness of a $K$-way neural network classifier $f : \mathbb{R}^d \to \mathbb{R}^K$ against a potential universal perturbation $\delta$ with $\ell_\infty$ norm constrained as $\|\delta\|_\infty \leq \epsilon$. In particular, we aim to certify and lower bound the worst-case accuracy of the neural network on $\{z^{(1)}, \ldots, z^{(n)}\}$ for any universal perturbation $\delta$ ($\|\delta\|_\infty \leq \epsilon$) applied to all the examples:

$$\min_{\|\delta\|_p \leq \epsilon} \frac{1}{n} \sum_{i=1}^n \mathbb{1}\left( \min_{j \neq y_i}\{\mathbf{m}_{y_i,j}(\mathbf{x}_i + \delta)\} > 0 \right), \tag{1}$$

where $\mathbf{m}_{y_i,j}(\mathbf{x}_i + \delta) = f_{y_i}(\mathbf{x}_i + \delta) - f_j(\mathbf{x}_i + \delta)$ is the margin between the ground-truth class $y_i$ and an incorrect class $j \neq y_i$, and the indicator checks whether the margin is positive for any $j \neq y_i$ when a perturbation $\delta$ is added. It is NP-hard to exactly verify Eq. (1) even for $n = 1$ and a small ReLU network (Katz et al., 2017). Thus recent neural network verifiers usually compute a lower bound for the margin as $\underline{\mathbf{m}}_{y_i,j}(\mathbf{x}_i + \delta) \leq \mathbf{m}_{y_i,j}(\mathbf{x}_i + \delta)$, and then we can replace $\mathbf{m}$ in Eq. (1) with $\underline{\mathbf{m}}$ to lower bound Eq. (1) and this bound also serves as a lower bound for the robustness.

### 3.2 LINEAR PERTURBATION ANALYSIS W.R.T. A UNIVERSAL PERTURBATION

We adopt linear relaxation based perturbation analysis (LiRPA) from previous works which focused on sample-wise perturbations, "auto_LiRPA" (Xu et al., 2020a) specifically, to obtain lower bounds on $\mathbf{m}_{y_i,j}(\mathbf{x}_i + \delta)$ represented as linear functions w.r.t. the universal perturbation $\delta$. but it is also feasible to use other verification frameworks such as Singh et al. (2019b); Wang et al. (2018b). auto_LiRPA can bound the output of a computational graph when its input nodes are perturbed, and it can produce linear functions w.r.t. the perturbed inputs nodes as linear bounds. Note that margin functions can be appended to the original neural classifier as the output of the computational graph, and thereby the margins can be bounded. When sample-wise perturbations are considered in previous works, the linear bounds can usually be written as

$$\forall i \in [n], \ \forall j \neq y_i, \ \forall \|\delta\|_\infty \leq \epsilon, \ \mathbf{m}_{y_i,j}(\mathbf{x}_i + \delta) \geq \underline{\tilde{\mathbf{a}}}_j^{(i)}(\mathbf{x}_i + \delta) + \underline{\tilde{\mathbf{b}}}_j^{(i)}, \tag{2}$$

where $\underline{\tilde{\mathbf{a}}}_j^{(i)}$ and $\underline{\tilde{\mathbf{b}}}_j^{(i)}$ are coefficients and biases in the linear bounds. This is achieved by relaxing nonlinear functions such as activation functions in the network with linear bounds and propagating linear coefficients through the computational graph. The right-hand-side (RHS) of Eq. (2) is a linear function w.r.t. $(\mathbf{x}_i + \delta)$. To obtain a final bound represented as a concrete number without relying on the $\delta$ variable, a *concretization* step can be applied on the RHS given the constraint on $\|\delta\|_\infty$, which eliminates the $\delta$ variable and lower bounds the RHS as $\underline{\tilde{\mathbf{a}}}_j^{(i)}(\mathbf{x}_i + \delta) + \underline{\tilde{\mathbf{b}}}_j^{(i)} \geq -\epsilon\|\underline{\tilde{\mathbf{a}}}_j^{(i)}\|_1 + \underline{\tilde{\mathbf{a}}}_j^{(i)}\mathbf{x}_i + \underline{\tilde{\mathbf{b}}}_j^{(i)}$.

However, the aforementioned concretization step considers the worst-case $\delta$ for each sample independently but a universal perturbation $\delta$ should be shared across all the examples. Thereby it will produce relatively loose and over-conservative results under the universal perturbation setting, as the perturbations are much stronger when each example can take an independent perturbation respectively compared to a single and universal perturbation for all the examples.

In contrast, we propose to obtain a tighter certification for universal perturbation. Unlike Eq. (2), we use auto_LiRPA to compute the linear lower bound with respect to $\delta$ instead of $(\mathbf{x}_i + \delta)$ by treating $\delta$ as a perturbed input node and $\mathbf{x}_i$ as a fixed input node in the computational graph:

$$\forall i \in [n], \ \forall j \neq y_i, \ \forall \|\delta\|_\infty \leq \epsilon, \ \ \mathbf{m}_{y_i,j}(\mathbf{x}_i + \delta) \geq \underline{\mathbf{a}}_j^{(i)}\delta + \underline{\mathbf{b}}_j^{(i)}, \tag{3}$$

where $\underline{\mathbf{a}}_j^{(i)}$ and $\underline{\mathbf{b}}_j^{(i)}$ are new coefficients and biases in the linear bound, and $\mathbf{x}_i$ does not appear on the RHS as it is fixed. In the next section, we will lower bound the worst-case accuracy Eq. (1) by solving an MILP problem based on Eq. (3).

### 3.3 An MILP Formulation to Lower Bound the Worst-Case Accuracy

In this section, we use linear bounds in Eq. (3) to compute a lower bound for the worst-case accuracy in Eq. (1). Specifically, by replacing each $\mathbf{m}_{y_i,j}$ in Eq. (1) with its lower bound from Eq. (3), we lower bound Eq. (1) by solving the following problem:

$$\text{minimize} \quad \frac{1}{n}\sum_{i=1}^{n} \mathbb{1}\left(\min_{j \neq y_i}\left\{\underline{\mathbf{a}}_j^{(i)}\delta + \underline{\mathbf{b}}_j^{(i)}\right\} > 0\right) \quad \text{s.t.} \quad \|\delta\|_\infty \leq \epsilon. \tag{4}$$

Now, we show that Eq. (4) can be rewritten into an MILP formulation:

**Theorem 1.** *Problem Eq. (4) is equivalent to the following MILP problem:*

$$\textit{minimize} \quad \vartheta$$

$$\textit{s.t.} \quad \vartheta = \frac{1}{n}\sum_{i=1}^{n} q^{(i)}, \tag{5}$$

$$\forall i \in [n], \ q^{(i)} \in \{0,1\}, \ -\tau(1-q^{(i)}) \leq \sum_{j \neq y_i}(\underline{\mathbf{a}}_j^{(i)}\delta + \underline{\mathbf{b}}_j^{(i)})s_j^{(i)} \leq \tau q^{(i)}, \tag{6}$$

$$\forall i \in [n], \forall j \neq y_i, \ s_j^{(i)} \in \{0,1\}, \ \sum_{j \neq y_i} s_j^{(i)} = 1, \tag{7}$$

$$\forall i \in [n], \forall j_1 \neq y_i, \forall j_2 \neq y_i, \ (\underline{\mathbf{a}}_{j_1}^{(i)}\delta + \underline{\mathbf{b}}_{j_1}^{(i)})s_{j_1}^{(i)} - \tau(1-s_{j_1}^{(i)}) \leq (\underline{\mathbf{a}}_{j_2}^{(i)}\delta + \underline{\mathbf{b}}_{j_2}^{(i)}), \tag{8}$$

$$\|\delta\|_\infty \leq \epsilon,$$

*where* $\tau \geq \max_{i \in [n]} \sum_{j \neq y_i} |\underline{\mathbf{a}}_j^{(i)}\delta + \underline{\mathbf{b}}_j^{(i)}|$ *is a sufficient large constant.*

In Theorem 1, given a universal perturbation $\delta$, for the $i$-th example, integer variable $q^{(i)} \in \{0,1\}$ denotes whether the the model is certifiably correct on this example based on linear bounds from Eq. (3), and the certified accuracy on the whole batch can be computed as Eq. (5). The model is certifiably correct on the $i$-th example when $\mathbf{m}_{y_i,j}(\mathbf{x}_i + \delta) \geq \underline{\mathbf{a}}_j^{(i)}\delta + \underline{\mathbf{b}}_j^{(i)} > 0$ holds for all $j \neq y_i$. We use an integer variable $s_j^{(i)} \in \{0,1\}$ to denote whether class $j$ is the hardest among all $j \neq y_i$ under the ceritification, i.e., $\forall j' \neq y_i, \underline{\mathbf{a}}_j^{(i)}\delta + \underline{\mathbf{b}}_j^{(i)} \leq \underline{\mathbf{a}}_{j'}^{(i)}\delta + \underline{\mathbf{b}}_{j'}^{(i)}$ holds, which is enforced by Eq. (8). We require each example to have exactly one hardest class $j$ with $s_j^{(i)} = 1$ (see Eq. (7)); in case that there are multiple classes with an equal lower bound on the margin function, it is valid to treat any of them as the hardest. Then we only need to check whether $\underline{\mathbf{a}}_j^{(i)}\delta + \underline{\mathbf{b}}_j^{(i)} > 0$ holds for the hardest class $j$ with $s_j^{(i)} = 1$, equivalently $\sum_{j \neq y_i}(\underline{\mathbf{a}}_j^{(i)}\delta + \underline{\mathbf{b}}_j^{(i)})s_j^{(i)} > 0$. In Eq. (6), as $\tau$ is sufficiently large, only $\sum_{j \neq y_i}(\underline{\mathbf{a}}_j^{(i)}\delta + \underline{\mathbf{b}}_j^{(i)})s_j^{(i)} \geq 0$ is effectively required when $q^{(i)} = 1$, and $\sum_{j \neq y_i}(\underline{\mathbf{a}}_j^{(i)}\delta + \underline{\mathbf{b}}_j^{(i)})s_j^{(i)} \leq 0$ is required when $q^{(i)} = 0$. Note that if exactly $\sum_{j \neq y_i}(\underline{\mathbf{a}}_j^{(i)}\delta + \underline{\mathbf{b}}_j^{(i)})s_j^{(i)} = 0$ happens, $q^{(i)} = 0$ will be taken by MILP due to the minimization objective, and thus it is still compatible with our goal

for checking $\underline{\mathbf{a}}_j^{(i)}\delta + \underline{\mathbf{b}}_j^{(i)} > 0$. Overall the MILP formulation minimizes the certified accuracy over all possible universal perturbation $\delta$ ($\|\delta\|_\infty \le \epsilon$), to finally produce a lower bound for Eq. (1). We formally prove this theorem in Appendix A.1, and we use Gurobi (Bixby, 2007) to solve the MILP.

Although it is possible to solve the whole certification algorithm through MILP (Tjeng et al., 2017), it will be computationally prohibitive. Even for very small networks with thousands of neurons, the number of integer variables in their MILP formulation will be proportional to the number of neurons. In contrast, by computing linear bounds first before solving MILP, the number of integer variables in our formulation is only proportional to the number of samples in a batch and the number of classes, and it does not depend on the size of the network, which makes it feasible in practice.

## 4 GENERALIZATION OF UNIVERSAL PERTURBATION

In the previous section, we proposed our robustness certification method against UPs. Note that the certification results are only guaranteed for the given batch of samples till now. In this section, we study how the certified accuracy computed on a batch approximates the certified accuracy computed on the entire data distribution.

Let $z^{(i)}$ be a random sample drawn from probability space $(\Omega, \mathcal{F}, \mathbb{P})$, which is endowed with a $\sigma$-algebra $\mathcal{F}$ and a probability measure $\mathbb{P}$. A dataset $\mathcal{D}_n \triangleq \{z^{(1)}, \ldots, z^{(n)}\}$ consists of $n$ observations drawn independently from $\Omega$ according to $\mathbb{P}$; equivalently it can be considered as a random point in $(\Omega^n, \mathcal{F}^n, \mathbb{P}^n)$, which is the $n$-fold Cartesian product of $\Omega$ equipped with the product $\sigma$-algebra $\mathcal{F}^n$ and the product $\mathbb{P}^n = \underbrace{\mathbb{P} \times \cdots \times \mathbb{P}}_{n \text{ times}}$. Let $\Delta$ denote the $l_\infty$ ball that contains all allowable perturbations $\Delta = \{\delta : \|\delta\|_\infty \le \epsilon\}$ with radius $\epsilon$. And let $\mathcal{B} : \Omega \to \mathbb{R}^{(d+1)K}$ be a linear bound generation procedure, and for each $z = (\mathbf{x}, y)$, it returns parameters $\{\underline{\mathbf{a}}_j, \underline{\mathbf{b}}_j\}_{j \ne y}$ of the linear lower bounds on the margins, i.e., $\mathbf{m}_{y,j}(\mathbf{x} + \delta) \ge \underline{\mathbf{a}}_j(\mathbf{x} + \delta) + \mathbf{b}_j$. In the proposed framework, $\mathcal{B}$ is instantiated to be auto_LiRPA (Xu et al., 2020a). Let $\mathcal{A}_n : \mathbb{R}^{(d+1)Kn} \to \Delta$ denote the MILP in Eq. (4), which return a perturbation $\delta$ given the linear bounds on the margins. The overall certification procedure is the composition of $\mathcal{A}_n$ and $\mathcal{B}$, denoted by $\mathcal{G} = \mathcal{A}_n \circ \underbrace{\mathcal{B} \circ \cdots \circ \mathcal{B}}_{n \text{ times}} \triangleq \mathcal{A}_n \circ \mathcal{B}^{\circ n}$.

For every data sample $z = (\mathbf{x}, y) \in \Omega$, we define the set

$$\Delta_z^{\mathcal{B}} := \left\{ \delta \in \Delta : \mathbb{1}\left( \min_{j \ne y} \left\{ \underline{\mathbf{a}}_j \delta + \underline{\mathbf{b}}_j \right\} > 0 \right) \right\}$$

as the set of perturbations such that the margin between the ground-truth class and any other class is certifiably positive *according to the linear bounds provided by* $\mathcal{B}$, i.e., the model is *certifiably robust* to any perturbation in this set, but it is still possible for the model to be robust to a perturbation $\delta \notin \Delta_z$. Note that the dependence of the set on $\mathcal{B}$ has been made explicit because $\underline{\mathbf{a}}_j, \underline{\mathbf{b}}_j$ depend on $\mathcal{B}$. Similarly, we define the set

$$\tilde{\Delta}_z := \left\{ \delta \in \Delta : \mathbb{1}\left( \min_{j \ne y} \left\{ \mathbf{m}_{y,j}(\mathbf{x} + \delta) \right\} > 0 \right) \right\} \tag{9}$$

as the set of all perturbations that are incapable of fooling the given model $f$, i.e., the data $z$ is *actually robust* to any perturbation in this set. Note that $\tilde{\Delta}_z$ is a superset of $\Delta_z^{\mathcal{B}}$, and unlike $\Delta_z^{\mathcal{B}}$, it does not depend on the linear bound generation procedure. We make the following definitions:

**Definition 1.** *The **certified robust probability** (CRP) of a given perturbation $\delta \in \Delta$ based on a linear bound generation procedure $\mathcal{B}$ is defined as*

$$V^{\mathcal{B}}(\delta) \triangleq \mathbb{P}(z \in \Omega : \delta \in \Delta_z^{\mathcal{B}}). \tag{10}$$

*The **actual robust probability** (ARP) of a given perturbation $\delta \in \Delta$ is defined as*

$$U(\delta) \triangleq \mathbb{P}(z \in \Omega : \delta \in \tilde{\Delta}_z). \tag{11}$$

*The **certified robust rate** (CRR) of a perturbation $\delta \in \Delta$ on an evaluation dataset $D_n$ based on a linear bound generation procedure $\mathcal{B}$ is*

$$\hat{V}^{\mathcal{B}}(\delta; D_n) \triangleq \frac{1}{n} \sum_{z \in D_n} \mathbb{1}(\delta \in \Delta_z^{\mathcal{B}}). \tag{12}$$

Equivalently, we can write the objective of Eq. (4) as $\min_{\delta \in \Delta} \hat{V}^{\mathcal{B}}(\delta; D_n)$. $\Delta_z^{\mathcal{B}}$ can be equivalently defined by the existence of a binary variable as in the MILP formulation in Theorem 1 and is thus nonconvex in general. In the following, we use $\hat{V}^{\mathcal{B}}(\delta)$ for $\hat{V}^{\mathcal{B}}(\delta; \mathcal{D}_n)$ if the evaluation dataset is $\mathcal{D}_n$ for notational simplicity. Note that $V^{\mathcal{B}}(\delta) \leq U(\delta)$ for any $\delta \in \Delta$ and the equality is attained when the equality in (3) is attained, i.e., the lower bound generated by $\mathcal{B}$ exactly matches the actual margin at any $\delta$. Now we present the following theorem that estimates the value of ARP based on the CRR computed from a batch of random samples.

**Theorem 2** (($1 - \xi$)-probable certification for ARP). *Given $\mathcal{G} = \mathcal{A}_n \circ \mathcal{B}^{\circ n}$ and $0 < \xi < 1$, for any $\delta$, it holds that*

$$\mathbb{P}^n \left( U(\delta) \geq \min_{\delta \in \Delta} \hat{V}^{\mathcal{B}}(\delta; D_n) + U(\delta^*) - V^{\mathcal{B}}(\delta^*) - t^*(\xi, n) \right) \geq 1 - \xi, \tag{13}$$

*where $t^*(\xi, n)$ is the root of the equation $(1 + 4t) \ln(1 + 4t) - 4t = \frac{4}{n} \ln(1/\xi)$ and $t^*(\xi, n)$ is a monotonically decreasing function in $n$ and $\xi$. $\delta^* = \arg \min_{\delta} U(\delta)$. Moreover, we have that*

$$\mathbb{P}^n \left( U(\delta) \geq \min_{\delta \in \Delta} \hat{V}^{\mathcal{B}}(\Delta; D_n) - t^*(\xi, n) \right) \geq 1 - \xi, \tag{14}$$

The proof can be found in Appendix A.2. Both bounds are interesting to interpret. The bound Eq. (13) shows that the discrepancy between the ARP of any perturbation and the CRR (i.e., the certified accuracy on a random batch) depends on $U(\delta^*) - V^{\mathcal{B}}(\delta^*)$ and $t^*(\xi, n)$. Given the trained model and the underlying data distribution, $\delta^*$ is fixed; hence, the term $U(\delta^*) - V^{\mathcal{B}}(\delta^*)$ depends on the tightness of linear bounds produced by $\mathcal{B}$. The tighter bounds $\mathcal{B}$ can provide, the smaller difference there will be between $U(\delta^*)$

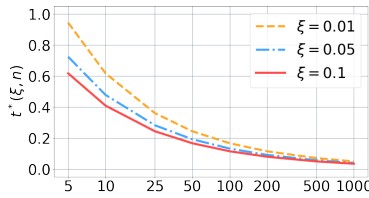

Figure 1: $t^*(\xi, n)$ vs. $n$.

and $V^{\mathcal{B}}(\delta^*)$. This bound suggests that plugging tighter linear bound generation techniques into our certification framework can potentially give rise to better approximation error. It is also interesting to note that the approximation error of the proposed certification framework $\mathcal{G} = \mathcal{A}_n \circ \mathcal{B}^{\circ n}$ exclusively depends on $\mathcal{B}$, not $\mathcal{A}_n$. This is because $\mathcal{A}_n$ always returns the optimal solution to the MILP, thereby not introducing any additional error. The second term $t^*(\xi, n)$ depends on the number of samples for certification and it vanishes as the $n$ grows (illustrated in Figure 1). The second bound (Eq. (14)) utilizes the fact that $U(\delta^*) - V^{\mathcal{B}}(\delta^*) \geq 0$, and is more relaxed but more convenient than the first bound (Eq. (13)) because the lower bound of the ARP can be calculated given the certification results on a batch, the number of samples in the batch, and the confidence level $1 - \xi$. In the Section 5.2, we will showcase the estimation of the ARP using this bound.

## 5 EXPERIMENT

### 5.1 EXPERIMENTAL SETUP

For evaluating the certification, we consider two benchmark datasets, MNIST (LeCun et al., 1998) and CIFAR-10 (Krizhevsky et al., 2009), widely adopted in existing works. We adopt 5 model structures from existing works (Singh et al., 2019a; Tjandraatmadja et al., 2020; Wang et al., 2021; Müller et al., 2022; Zhang et al., 2022a): Conv-small, Conv-4-layer, Conv-big on MNIST, and ResNet-2B, ResNet-4B on CIFAR-10, with details in Appendix B. We use the CRR and attack-ACC (accuracy under an attack) as the metrics. All the results are averaged over three runs with different random seeds on 100 random samples for each dataset. Further experimental details can be found in Appendix B.

### 5.2 EVALUATION

**Comparing to existing robustness certification** We first focus on evaluating our certification results compared to existing robustness certification for sample-wise perturbation. There are several competitive frameworks such as Singh et al. (2019b); Bunel et al. (2020); Henriksen & Lomuscio (2021), and we compare with auto_LiRPA (Xu et al., 2020a) specifically for its state-of-the-art

Table 1: Certification results on naturally trained MNIST models (CRR %).

| | Conv-small | | Conv-4-layer | | Conv-big | |
|---|---|---|---|---|---|---|
| | Sample-Wise | Ours | Sample-Wise | Ours | Sample-Wise | Ours |
| $\epsilon = 4/255$ | 94.00±3.74 | 95.00±2.94 | 93.33±2.87 | 95.33±1.25 | 41.67±8.34 | 45.33±6.60 |
| $\epsilon = 6/255$ | 80.67±6.18 | 86.33±4.50 | 69.00±5.89 | 78.67±3.68 | 6.67±2.49 | 8.67±3.09 |
| $\epsilon = 8/255$ | 52.33±6.24 | 63.33±5.25 | 40.67±6.13 | 52.0±6.68 | 0.00±0.00 | 0.00±0.00 |

Table 2: Certification results on PGD-32 adversarially trained MNIST models (CRR %).

| | Conv-small | | Conv-4-layer | | Conv-big | |
|---|---|---|---|---|---|---|
| | Sample-Wise | Ours | Sample-Wise | Ours | Sample-Wise | Ours |
| $\epsilon = 16/255$ | 94.67±1.25 | 95.33±1.70 | 96.67±0.47 | 97.00±0.00 | 69.67±1.70 | 70.67±1.70 |
| $\epsilon = 32/255$ | 34.67±5.31 | 44.33±9.39 | 54.33±0.47 | 67.0±2.16 | 0±0 | 0±0 |

Table 3: Certification results on PGD-8 adversarially trained CIFAR-10 models (CRR %).

| | ResNet-2B | | ResNet-4B | |
|---|---|---|---|---|
| | Sample-Wise | Ours | Sample-Wise | Ours |
| $\epsilon = 1/255$ | 43.33±2.62 | 46.33±0.94 | 51.00±3.27 | 54.67±4.03 |
| $\epsilon = 3/255$ | 28.67±2.05 | 33.00±1.41 | 0.00±0.00 | 0.00±0.00 |

performance (Bak et al., 2021). We consider models from both natural training and adversarial training. For MNIST, we evaluate the two certification methods on models naturally trained, and PGD-32 (PGD with $\ell_\infty$-norm $\frac{32}{255}$) adversarially trained models (Madry et al., 2018). Table 1 details the results on naturally trained MNIST models. Our method provides much tighter bounds than sample-wise certification results across all the settings. Table 2 illustrates results on PGD-32 trained MNIST models. With adversarial training, we observe the certified robustness of the models against UPs also largely increased compared naturally trained models in Table 1. Our certification results are still much tighter than sample-wise results especially under settings with larger perturbations. On CIFAR-10, we only evaluate the results on adversarially trained models (PGD-8 for ResNet-2B and ResNet-4B) as the naturally trained model is severely susceptible to perturbations on CIFAR-10 (see Figure 5) even with $\epsilon = \frac{1}{255}$. To sum up, our method can provide tighter robustness certification than existing sample-wise methods.

**Estimation of the lower bound of ARP** Figure 2 illustrate the application of Theorem 2 with CRR. We use the naturally trained Conv-4-layer on MNIST as an example and we set $\epsilon = \frac{6}{255}$. We demonstrate the estimation of 0.9-probable certification for the lower bound of ARP, or $\underline{ARP}$, by setting $\xi = 0.1$. From Figure 2, we can learn that the empirical results, CRR, can be tighter when more samples are considered in certification. Incorporating more samples also makes the estimated $\underline{ARP}$ much closer to the CRR (as $t^*(\xi, n)$ is smaller). Such an observation shows that when incorporating more samples in certification, the empirical results would better reflect the actual robustness of the whole population.

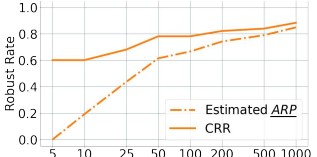

Figure 2: Different numbers of samples vs. the lower bound of ARP, i.e., $\underline{ARP}$, (Conv-4-layer, MNIST, at $\epsilon = \frac{6}{255}$, $\xi = 0.1$).

In particular, when using 1000 samples, the result can be interpreted as the ARP is larger than 84.73% with at least a 90% probability.

**Validating with UAP attacks** We then validate the robustness certification results with UAP attacks as CRR should lower bound the attack-ACCs. We consider three SOTA UAP attacks: Adv-UAP (Li et al., 2022), Cos-UAP (Zhang et al., 2021a), and DF-UAP (Zhang et al., 2020), detailed in in Appendix B. Figure 3 compares CRR and attack-ACCs on PGD-32 trained MNIST models. As shown in Figure 3, CRRs are indeed lower than all the attack-ACCs as expected.

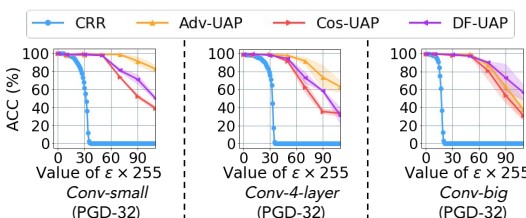

Figure 3: CRRs vs. UAP-attacked ACCs (MNIST, PGD-32 trained models).

**Validating with backdoor attacks** We also validate whether CRR still lower bounds the attack-ACCs in backdoor attacks. We consider two backdoor triggers namely a blended trigger (Chen et al., 2017) with a small $\ell_\infty$-norm ($\|\delta\|_\infty = \frac{5}{255}$, referred as the stealthy trigger), and the BadNets (Gu

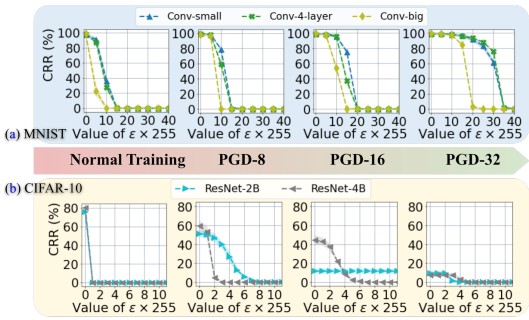

Figure 4: Visual examples of the backdoor poisoned samples, the triggers, and certified UP-ACCs (Conv-small, MNIST). We magnified the stealthy trigger by $51 \times$ for visualization.

et al., 2019) ($\|\delta\|_\infty = \frac{255}{255}$). All the attacks utilize the same poison ratio, 20% following existing works (Zeng et al., 2021). The visual example of the poisoned sample, the triggers, and the certification results are listed in Figure 4. Under the setting of the stealthy blended backdoor, we find that the CRR drops dramatically before reaching the trigger's norm ($\|\delta\|_\infty = \frac{5}{255}$) compared to the same model trained on clean MNIST. This observation verifies the correctness of CRR and its potential to reveal stealthy $l_\infty$-bounded backdoor attacks in the current trend of backdoor development with smaller $l_\infty$-norm constraints, e.g., Zeng et al. (2022b); Zhao et al. (2020). However, assuming an $\ell_p$ norm bound of the backdoor triggers is not widely accepted in traditional backdoor settings. Thus, we also present the results of BadNets (with $\|\delta\|_\infty = \frac{255}{255}$) in Figure 4. We consider the backdoor model trained from scratch or fine-tuned from the clean model. The CRR is still lower-bounding the attack's deployed $\ell_\infty$ bound of the trigger. However, as the trigger has a large $\ell_\infty$ norm, the CRRs of poisoned models are of no difference to the clean model and thus not that useful. Nevertheless, in Section 5.3, we show a simple twist of the certification framework to help reveal backdoors' existence.

## 5.3 IMPLICATIONS OF ROBUSTNESS CERTIFICATION AGAINST UPS

Now we explore the potential implications of our robustness certification against UPs. We focus on 3 case studies on model structure comparison, UAP defenses comparison, and backdoor detection.

**Comparing model structures** One implication of robustness certification regarding UPs is to compare different model structures and training strategies regarding the certified robustness against UPs. Figure 5 depicts the certification results of all the considered model structures with different training settings on MNIST and CIFAR-10. We consider both naturally trained and PGD trained models with different $l_\infty$ perturbation norm. In Figure 5 (a) on MNIST, we find that the largest model, Conv-big, shows the worst certified robustness against UPs. But the smallest Conv-small's CRR is higher than that of Conv-4-layer under naturally trained setting, PGD-8, and PGD-16, but not PGD-32. The only difference between Conv-small and Conv-4-layer is that Conv-4-layer

Figure 5: CRR comparison with different training settings and structures on (a) MNIST and (b) CIFAR-10. PGD-8, PGD-16, and PGD-32 stand for PGD training with $\ell_\infty$ norm $\frac{8}{255}, \frac{16}{255}, \frac{32}{255}$ respectively.

uses a larger padding step which resulting a slightly larger hidden layer (see Appendix B). Based on the observation, there is an interesting trade-off between model size and certified robustness against UPs: A slightly larger structure can help the model obtain better certified robustness when adopting adversarial training, potentially due to increased model capacity. Such an observation can be further illustrated in Figure 5 (b). Specifically, ResNet-2B's CRR would drop to random guessing when using PGD-16, while ResNet-4B can still maintain a certain scale of CRR. But even larger models Figure 5 (a) have worse certified robustness, potentially due to looser certified bounds.

**Implication to UAP defenses** Another implication of the CRR is to compare existing UAP defenses regarding their efficacy. We consider three types of defenses and five different defenses in total: FGSM and PGD sample-wise adversarial training (Goodfellow et al., 2014; Madry et al., 2017;

Table 4: UAP defenses at $|\delta|_\infty = \frac{16}{255}$ (ResNet-4B, CIFAR-10).

|  | **Normal** | **FGSM** | **PGD** | **UAT-FGSM** | **UAT-PGD** | **IBP** |
|---|---|---|---|---|---|---|
| **Ori-ACC** | 99.03 | 98.97 | 98.88 | 98.99 | 98.99 | 98.64 |
| **CRR** | 0.33±0.47 | 27.33±5.91 | 30.67±6.34 | 24.00±0.82 | 28.99±2.49 | 95.66±2.05 |
| **DF-UAP** | 98.94±0.04 | 98.91±0.01 | 98.84±0.02 | **98.82±0.08** | 98.94±0.04 | 98.65±0.02 |
| **Cos-UAP** | **98.62±0.15** | **98.87±0.06** | **98.82±0.02** | 98.88±0.08 | **98.89±0.02** | **98.59±0.03** |
| **Adv-UAP** | 98.96±0.04 | 98.94±0.03 | 98.86±0.02 | 98.89±0.02 | 98.96±0.00 | 98.68±0.02 |
| **Worst** | -0.41±0.15 | -0.10±0.06 | -0.06±0.02 | -0.17±0.08 | -0.10±0.02 | -0.05±0.03 |

Wong et al., 2019); universal adversarial training (UAT) with FGSM or PGD synthesizing UPs (Shafahi et al., 2020); sample-wise certified defense with Interval Bound Propagation (IBP) training (Gowal et al., 2018; Mirman et al., 2018). The defended models are further evaluated with UAP attacks and certification. The results with a small perturbation radius $|\delta|_\infty = \frac{16}{255}$ are shown in Table 4. Additional results with a larger perturbation radius ($|\delta|_\infty = \frac{80}{255}$) are in Table 7, Appendix C. We use the row titled "**Worst**" to record the maximum accuracy drop using UAP attacks compared to clean accuracy. Surprisingly, in Table 4, we find the CRR of models trained with UAT is worse than their sample-wise adversarial training counterparts (i.e., UAT-PGD results are worse than PGD). However, in the case of larger perturbation radius (Table 7, Appendix C), the UAT-trained models can achieve higher CRR than the sample-wise counterparts. Such an observation indicates an under-explored trade-off between perturbation radius and UAP defense method on CRR. The CRR result from the IBP-trained model is much tighter than others, as IBP directly optimizes over an objective for certified robustness and tightens the certified bounds for all the neurons. Moreover, CRR is also aligned with the worst attacked accuracy drop and can be an indicator for comparing different settings of UAP defenses.

**Implication to backdoor defenses** We evaluated the effectiveness of the CRR in revealing potential backdoors in the above section, but the effectiveness is yet only limited to triggers with small perturbations. This section presents a simple twist on the certification framework by teaming up with adversarial training (PGD-16). We depict the average class-wise certification results on 10 ResNet-4B models trained with different random seeds over different BadNets poison ratios in Figure 4. Based on the re-

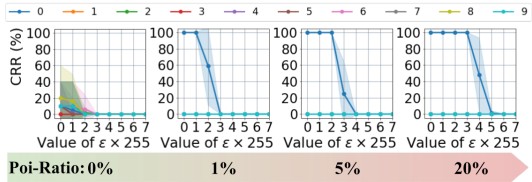

Figure 6: CRRs on adversarial-trained models over CIFAR-10 datasets that contains a different ratio of poisons (**BadNets**, target label is 0). Different class indexes are depicted with different colors listed a the top.

sults, we find the certification can reliably reveal the targeted label and justify how mighty the backdoor attack is (i.e., the CRR is aligned with the poison ratio used). Additional results on the Smooth attack (Zeng et al., 2021) and $\ell_2$ invisible attack (Li et al., 2020a) are listed in Appendix C, which share similar observations. The reason of the successful identification is that, naturally, the adversarial training would force the model to learn more from the reliable features and thus make standard backdoors stand out from benign features of the data (i.e., easier to be learned by the model), as also discussed in Weng et al. (2020). Thus after training a model with adversarial training with large perturbation radius, the model would likely engrave the trigger and thus have a high CRR only on the target label. CRR by our proposed method provides an intriguing point of view to determine the attack's strength (i.e., poison ratio).

## 6 CONCLUSION

In this work, we present the first focused study on certifying neural networks' robustness against UPs. In contrast to previous robustness certification works that focused on sample-wise perturbations, we formulate the certification problem against UPs by emphasizing sharing a universal perturbation between different samples. We propose a combination of linear relaxation-based bounds and MILP to solve the problem. We also present a theoretical analysis framework to estimate the certification result for the entire population based on results from a batch of random samples. Extensive experiments reveal that our certification imposes tighter results than directly applying existing sample-wise robustness certifications. In addition, we discuss and demonstrate how robustness certification against UPs could facilitate comparing certified robustness between different model structures and defense methods and provide reliable backdoor detection.

ACKNOWLEDGEMENT

This work is partially funded by Sony AI. This work is also supported in part by NSF under IIS-2008173, IIS-2048280, and by Army Research Laboratory under W911NF-20-2-0158. RJ and the ReDS lab appreciate the support of The Amazon - Virginia Tech Initiative for Efficient and Robust Machine Learning and the Cisco Award. YZ and ZS are both supported by the Amazon Fellowship.

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

Table 5: Table of Notations.

| Notation | Definition | | Notation | Definition |
|---|---|---|---|---|
| $d$ | dimensionality of the input vector | | $\Omega$ | space of all possible data, $z \in \Omega$. |
| $K$ | number of output classes | | $\delta \in \Delta$ | universal perturbation, $\Delta$ is the space satisfies the norm-constrains |
| $f : \mathbb{R}^d \to \mathbb{R}^K$ | neural network classifier | | $\|\delta\|_p$ | $\ell_p$ norm of the universal perturbation, $p \geq 1$ |
| $z^{(i)} = (\mathbf{x}_i, y_i)$ | clean data sample with index $i$ | | $\epsilon$ | Norm-constraint on the universal perturbation, i.e., $\|\delta\|_p \leq \epsilon$ |
| $\mathbf{x}_i \in \mathbb{R}^d$ | sample $z^{(i)}$'s input feature | | $\underline{\mathbf{a}}_j^{(i)} \in \mathbb{R}^d$ | coefficients of the linear lower bound of the margin from $y_i$ to $j$ |
| $y_i \in \mathbb{R}$ | sample $z^{(i)}$'s label, $y_i \in \{1, \ldots, K\}$ | | $\underline{\mathbf{b}}_j^{(i)} \in \mathbb{R}$ | bias term of the linear lower bound of margin from $y_i$ to $j$ |
| $\mathcal{A}(\cdot) : \Omega \to \Delta$ | certification procedure, returns a worst-case $\delta$ | | $\tilde{\Delta}_z$ | set of perturbations incapable of fooling the classifier at sample $z$ |

# A    PROOFS

## A.1    PROOF OF THEOREM 1

Let $\hat{\vartheta}$ be the solution of the MILP problem in the theorem, and let $\tilde{\vartheta}$ be the solution to Eq. (4). Theorem 1 states that $\hat{\vartheta} = \tilde{\vartheta}$. We formally prove the equivalence below.

*Proof.* We first show that $\hat{\vartheta} \leq \tilde{\vartheta}$. In Eq. (4), there exists some $\tilde{\delta}$ such that

$$\tilde{\delta} = \arg\min_{\|\delta\|_\infty \leq \epsilon} \frac{1}{n} \sum_{i=1}^{n} \mathbb{1}\left( \min_{j \neq y_i} \left\{ \underline{\mathbf{a}}_j^{(i)} \delta + \underline{\mathbf{b}}_j^{(i)} \right\} > 0 \right).$$

Then, for every $i \in [n]$, take the following values for variables in the MILP formulation:

$$q^{(i)} = \mathbb{1}\left( \min_{j \neq y_i} \left\{ \underline{\mathbf{a}}_j^{(i)} \tilde{\delta} + \underline{\mathbf{b}}_j^{(i)} \right\} > 0 \right),$$

$$\vartheta = \tilde{\vartheta} = \frac{1}{n} \sum_{i=1}^{n} q^{(i)},$$

$$\forall j \neq y_i, \ s_j^{(i)} = \mathbb{1}(j = j'), \ \text{ where } \ j' = \arg\min_{j \neq y_i} \underline{\mathbf{a}}_j^{(i)} \tilde{\delta} + \underline{\mathbf{b}}_j^{(i)},$$

and it is easy to see that the values for these variables satisfy all the constraints in the MILP problem. Thus the result of the minimization in the MILP should be no smaller than $\tilde{\vartheta}$, i.e., $\hat{\vartheta} \leq \tilde{\vartheta}$.

We now show that $\tilde{\vartheta} \leq \hat{\vartheta}$. We use $\hat{\delta}, \hat{q}, \hat{s}$ to denote the values of $\delta, q, s$ variable in the solution of MILP. For every $i \in [n]$, Eq. (7) ensures that there exists exactly one $\hat{j}$ $(\hat{j} \neq y_i)$ with $\hat{s}_{\hat{j}}^{(i)} = 1$, and Eq. (8) ensures that for all $j \neq y_i$, $\underline{\mathbf{a}}_{\hat{j}}^{(i)} \hat{\delta} + \underline{\mathbf{b}}_{\hat{j}}^{(i)} \leq \underline{\mathbf{a}}_j^{(i)} \hat{\delta} + \underline{\mathbf{b}}_j^{(i)}$ holds. Thus

$$\sum_{j \neq y_i} (\underline{\mathbf{a}}_j^{(i)} \delta + \underline{\mathbf{b}}_j^{(i)}) \hat{s}_j^{(i)} = \min_{j \neq y_i} \{ \underline{\mathbf{a}}_j^{(i)} \hat{\delta} + \underline{\mathbf{b}}_j^{(i)} \}.$$

According to Eq. (6), if $\hat{q}^{(i)} = 1$, $\sum_{j \neq y_i} (\underline{\mathbf{a}}_j^{(i)} \delta + \underline{\mathbf{b}}_j^{(i)}) \hat{s}_j^{(i)} \geq 0$ holds. In case that $\sum_{j \neq y_i} (\underline{\mathbf{a}}_j^{(i)} \delta + \underline{\mathbf{b}}_j^{(i)}) \hat{s}_j^{(i)} = 0$, Eq. (6) also holds with $\hat{q}^{(i)} = 0$, and due to the minimization objective of MILP, $\hat{q}^{(i)} = 0$ instead of $\hat{q}^{(i)} = 1$ will be taken. Thus $\sum_{j \neq y_i} (\underline{\mathbf{a}}_j^{(i)} \delta + \underline{\mathbf{b}}_j^{(i)}) \hat{s}_j^{(i)} > 0$ strictly holds when $\hat{q}^{(i)} = 1$. And if $\hat{q}^{(i)} = 0$, $\sum_{j \neq y_i} (\underline{\mathbf{a}}_j^{(i)} \delta + \underline{\mathbf{b}}_j^{(i)}) \hat{s}_j^{(i)} \leq 0$ holds. Thus

$$\hat{q}^{(i)} = \mathbb{1}\left( \sum_{j \neq y_i} (\underline{\mathbf{a}}_j^{(i)} \delta + \underline{\mathbf{b}}_j^{(i)}) \hat{s}_j^{(i)} > 0 \right) = \mathbb{1}\left( \min_{j \neq y_i} \{ \underline{\mathbf{a}}_j^{(i)} \hat{\delta} + \underline{\mathbf{b}}_j^{(i)} \} > 0 \right).$$

Thereby

$$\hat{\vartheta} = \frac{1}{n} \sum_{i=1}^{n} \hat{q}^{(i)} = \frac{1}{n} \sum_{i=1}^{n} \mathbb{1}\left( \min_{j \neq y_i} \left\{ \underline{\mathbf{a}}_j^{(i)} \hat{\delta} + \underline{\mathbf{b}}_j^{(i)} \right\} > 0 \right),$$

and then the result of Eq. (4) is no smaller than $\hat{\vartheta}$, i.e., $\tilde{\vartheta} \leq \hat{\vartheta}$.

Hence $\hat{\vartheta} = \tilde{\vartheta}$ is proved. □

## A.2 PROOF OF THEOREM 2

Let $\delta^* = \arg\min_{\delta \in \Delta} U(\delta)$ be the optimal universal perturbation that minimizes the ARP, let $\overline{\delta}_n$ be the value returned by $\mathcal{G}(\mathcal{D}_n)$, and let $\tilde{\delta} = \arg\min_{\delta \in \Delta} V^{\mathcal{B}}(\delta)$ that minimizes the CRP. We introduce the following lemma:

**Lemma 1.** *Given $\mathcal{A}_n$, it holds that*

$$\mathbb{P}^n(\hat{V}^{\mathcal{B}}(\delta^*) - V^{\mathcal{B}}(\delta^*) > t^*(\xi, n)) \le \xi \tag{15}$$

*where $t^*(\xi, n)$ is the root of the equation $(1 + 4t)\ln(1 + 4t) - 4t = \frac{4}{n}\ln(1/\xi)$.*

*Proof.* Let $q^{(i)} = \mathbb{1}(\delta^* \in \Delta_{z^{(i)}}^{\mathcal{B}})$ which can also be interpreted as $\mathbb{1}\left(\min_{j \ne y_i}\left\{\underline{\mathbf{a}}_j^{(i)}\delta^* + \underline{\mathbf{b}}_j^{(i)}\right\} > 0\right)$. Then, $\hat{V}^{\mathcal{B}}(\delta^*) = \frac{1}{n}\sum_{i=1}^n q^{(i)}$ and $V^{\mathcal{B}}(\delta^*) = \mathbb{E}[\frac{1}{n}\sum_{i=1}^n q^{(i)}] = \mathbb{E}[q^{(i)}]$. Let $\sigma^2$ denote the variance of $q^{(i)}$. Since $q^{(i)}$ is a binary random variable, we have that $\sigma^2 \le 1/4$. Let $h(u) = (1 + u)\ln(1 + u) - u$. For any $t > 0$, we have that

$$\mathbb{P}^n(\frac{1}{n}\sum_{i=1}^n q^{(i)} - \mathbb{E}[q^{(i)}] > t) \le \exp\left(-n\sigma^2 h(\frac{t}{\sigma^2})\right) \tag{16}$$

$$\le \exp\left(-\frac{n}{4}h(4t)\right), \tag{17}$$

where the first inequality is a direct application of the Bennett's inequality (Bennett, 1962), and the second inequality is due to the fact that $n\sigma^2 h(\frac{t}{\sigma^2})$ is a monotonically decreasing function of $\sigma^2$. Let $t(\epsilon, n)$ denote the root of $\exp\left(-\frac{n}{4}h(4t)\right) = \xi$. Then, it follows that $\mathbb{P}^n(\frac{1}{n}\sum_{i=1}^n q^{(i)} - \mathbb{E}[q^{(i)}] > t(\epsilon, n)) \le \xi$. $\square$

Then we prove Theorem 2 to certify the robustness of a classifier against the worst-case attack $\delta^*$.

*Proof.* We use the following relations: for any $\delta \in \Delta$,

$$\begin{aligned}
U(\delta) &\ge \min_{\delta \in \Delta} U(\delta) \\
&= U(\delta^*) \\
&= \underbrace{U(\delta^*) - V^{\mathcal{B}}(\delta^*)}_{(i)} + \underbrace{V^{\mathcal{B}}(\delta^*) - \hat{V}^{\mathcal{B}}(\delta^*)}_{(ii)} + \underbrace{\hat{V}^{\mathcal{B}}(\delta^*) - \hat{V}^{\mathcal{B}}(\overline{\delta}_n)}_{(iii)} + \hat{V}^{\mathcal{B}}(\overline{\delta}_n) \\
&\ge (i) + (ii) + \hat{V}^{\mathcal{B}}(\overline{\delta}_n),
\end{aligned} \tag{18}$$

where $(ii)$ can be bounded by applying the concentration inequality in Lemma 1; $(iii) \ge 0$ due to the optimality of $\overline{\delta}_n = \arg\min_{\delta \in \Delta} \hat{V}^{\mathcal{B}}(\delta)$. Combining these bounds yields Theorem 2. $\square$

## B FURTHER DETAILS ON EXPERIMENTAL SETTINGS

We use one server equipped with a total of 8 RTX A6000 GPUs as the hardware platform. PyTorch (Paszke et al., 2019) is adopted as the implementation framework. We detail the model structures used in our experiment in Table 6. All of the model structures used in this work were also considered in other existing robustness certification works as the standard set-ups: Conv-small, Conv-4-layer, Conv-big on the MNIST (Singh et al., 2019a; Tjandraatmadja et al., 2020; Wang et al., 2021; Müller et al., 2022), and ResNet-2B, ResNet-4B on the CIFAR-10 (Zhang et al., 2022a). We use Adadelta (Zeiler, 2012) as the optimizer with a learning rate set to 0.1 for all the model training process (including the adversarial training for the model updating step as well). For MNIST models, we train each model with 60 epochs. For CIFAR-10 models, we train each model with 500 epochs to ensure full convergence. For adversarial training adopted in the main text, the number of steps in PGD attacks is 7; step-size for PGD is set as $\frac{\epsilon}{4}$. For IBP training, we use the implementation in Shi et al. (2021).

Table 6: Model structures in our experiments. $\text{Conv}(1, 16, 4)$ stands for a conventional layer with one input channel, 16 output channels, and a kernel size of $4 \times 4$. $\text{Linear}(1568, 100)$ stands for a fully connected layer with 1568 input features and 100 output features. $\text{ResBlock}(16, 32)$ stands for a residual block with 16 input channels and 32 output channels. ReLU activation function is adopted between any two consecutive layers.

| Model name | Model structure |
|---|---|
| Conv-small (MNIST) | $\text{Conv}(1, 16, 4) - \text{Conv}(16, 32, 4) - \text{Linear}(800, 100) - \text{Linear}(100, 10)$ |
| Conv-4-layer (MNIST) | $\text{Conv}(1, 16, 4) - \text{Conv}(16, 32, 4) - \text{Linear}(1568, 100) - \text{Linear}(100, 10)$ |
| Conv-big (MNIST) | $\text{Conv}(1, 32, 3) - \text{Conv}(32, 32, 4) - \text{Conv}(32, 64, 3) - \text{Conv}(64, 64, 4) - \text{Linear}(3136, 512) -$ $\text{Linear}(512, 512) - \text{Linear}(512, 10)$ |
| ResNet-2B (CIFAR-10) | $\text{Conv}(3, 8, 3) - \text{ResBlock}(8, 16) - \text{ResBlock}(16, 16)) - \text{Linear}(1024, 100) - \text{Linear}(100, 10)$ |
| ResNet-4B (CIFAR-10) | $\text{Conv}(3, 16, 3) - \text{ResBlock}(16, 32) - \text{ResBlock}(32, 32)) - \text{Linear}(512, 100) - \text{Linear}(100, 10)$ |

Now, we details the UAP attacks considered in the experiment for validating the certification results, namely the Adv-UAP (Li et al., 2022), Cos-UAP (Zhang et al., 2021a), and DF-UAP (Zhang et al., 2020). The design of each UAP attack's synthesis procedure distinguishes these attacks. Specifically, Adv-UAP synthesizes and generates adversarial examples for each input before synthesizing the UAP, which has shown to be more effective in finding stronger UAPs. Cos-UAP produces UAP by reducing the Cosine similarity between the original output logits and the disturbed logits; DF-UAP employs a similar loss as listed in the C&W attack (Carlini & Wagner, 2017), which aims to reduce the distance between the ground-truth label's logits and the maximum logits of the rest.

Now we provide the detailed settings of the backdoor target-class identification in Section 5.3. For the threat model, we consider the scenario where the defender aims to determine if a backdoor attack resides in a given dataset, identify the target class, and justify how potent the attack is if there is an identified attack. We assumes the defender has access to the training set to be inspected, with no additional clean validation data required. To conduct the instantiated case shown in Section 5.3, the defender adversarially trains (with PGD-16) 10 different models on the inspecting training dataset and obtaining the averaging CRR results in a class-wise manner. Especially as we assume no additional clean validation data is required, we pass through 100 random noise into the certifying models to obtain the results in Figure 6, 7, 8.

## C ADDITIONAL RESULTS

### C.1 ADDITIONAL RESULTS ON UAP DEFENSES COMPARISON

Table 7 details the results of UAP defenses comparison under the large-norm setting ($\epsilon = \frac{80}{255}$). Noting all the defenses adopted are also incorporated with the same expense. For large-norm settings, we find that only the certified-robustness training ends up with a CRR larger than 0. Apart from its actual effectiveness, as mentioned in the main text, the IBP-trained model also ends up with much tighter intermediate linear bounds (i.e., $\underline{\mathbf{a}}$ and $\underline{\mathbf{b}}$ are tighter). Even though our work can only return a positive CRR on the IBP-trained model, the certification results are still aligned with the actual attack results, as the IBP-trained model would have stronger robustness than the other models in terms of the least change in the ACC drop.

Table 7: UAP defenses at $|\delta|_\infty = \frac{80}{255}$ (ResNet-4B, CIFAR-10).

| | Normal | FGSM | PGD | UAT-FGSM | UAT-PGD | IBP |
|---|---|---|---|---|---|---|
| Ori-ACC | 99.03 | 98.94 | 98.75 | 99.20 | 99.02 | 96.04 |
| CRR | 0 | 0 | 0 | 0 | 0 | 40.67±4.11 |
| DF-UAP | **26.46±9.27** | 98.10±0.02 | 97.40±0.05 | 98.56±0.07 | 98.45±0.02 | 94.89±0.05 |
| Cos-UAP | 41.80±16.27 | **79.22±1.41** | **95.13±0.37** | **96.68±0.12** | **97.50±0.11** | 95.18±0.07 |
| Adv-UAP | 37.69±7.51 | 97.93±0.10 | 97.36±0.03 | 98.40±0.05 | 98.54±0.02 | **94.83±0.05** |
| Worst | -72.57±9.27 | -19.72±1.41 | -3.62±0.37 | -2.52±0.12 | -1.52±0.11 | -1.21±0.05 |

### C.2 ADDITIONAL RESULTS ON BACKDOOR TARGET-CLASS IDENTIFICATION

We now provide additional results on implementing the certification framework to identify the existence of backdoor attacks. In this section, the results provided are evaluated against the Smooth attack (Zeng et al., 2021) and the $l_2$-invisible attack (Li et al., 2020a). Figure 7,8 illustrate the results on the Smooth attack and $l_2$-invisble attack respecively. Based on the results, we find the certification can also reliably reveal the targeted label and justify how powerful the backdoor attack is for Smooth attack and the $l_2$-invisible attack.

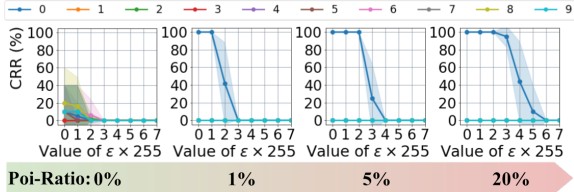

Figure 7: Certified UP-ACCs on adversarial-trained models over CIFAR-10 datasets that contains a different ratio of poisons (**Smooth attack** with the target label being set to 0).Different class indexes are depicted with different colors listed a the top.

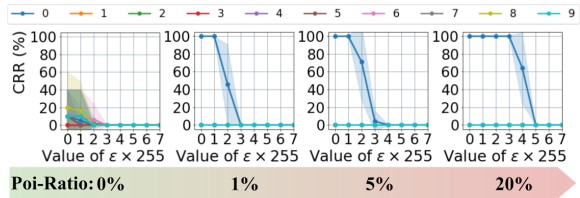

Figure 8: Certified UP-ACCs on adversarial-trained models over CIFAR-10 datasets that contains a different ratio of poisons ($l_2$-**inv** attack with the target label being set to 0).Different class indexes are depicted with different colors listed a the top.

# D  BROADEN IMPACT AND LIMITATIONS

## D.1  UAP AND BACKDOOR ATTACKS

UAP attacks aim to synthesize a UP by accessing and analyzing the output of a trained neural network. Backdoor attacks aim to insert a predefined trigger into the neural network and ensure an effective attack without accessing and analyzing the output after the model is trained over the poisoned samples. Many existing works have found these two paralleled lines of work have interesting intersections. In particular, the formulation of UAP synthesizing has also inspired or has its interesting counterparts in backdoor attacks or defense designs. For example, Li et al. (2020a); Zhang et al. (2021b) designed their backdoor trigger via a similar process of synthesizing UAP using a trained model. Kolouri et al. (2020); Zeng et al. (2022a) adopted this interesting intersection between UAP and backdoor attacks to provide identification of backdoors or conduct online removal of backdoors. Suppose we view these two attack paradigms at the inference time (with a trained model). In that case, mitigation defenses and robustness synthesizing tools for both attacks can be developed for general robustness to UP.

## D.2  LIMITATIONS

**Unconstrained or Large $\ell_\infty$-norm Attacks:** Some of the UAP attacks are generated without specifying a constraint (Brown et al., 2017), and in most backdoor attacks, the trigger inserted does not have a constrained $\ell_\infty$ norm. If the attack can have an unconstrained $\ell_\infty$ or a very large $\ell_\infty$ norm, only trivial certification results can be obtained from our certification. This limitation also commonly exists in state-of-the-art sample-wise certification methods (Wang et al., 2021; Ferrari et al., 2021). In fact, any certification procedure requires some constraints on potential perturbations and does not apply to unbounded perturbations. This open problem calls for answers and the attention for future research.

**Computational Cost:** Supporting large models and large datasets can be computationally costly for our certification. Existing works for certifying pre-trained models (Wang et al., 2021; Ferrari et al., 2021) are also commonly limited to moderate-sized networks, and the cost of our method is lower bounded by existing linear bound propagation frameworks that we use to obtain the linear bounds before solving the MILP problem. It remains a challenging open-problem for scaling to larger-scale networks, such as models for ImageNet (Deng et al., 2009).

