# OpenReview forum: "Towards Robustness Certification Against Universal Perturbations"
_ICLR.cc/2023/Conference — ICLR 2023 poster_

### Official Review · Reviewer_GEmp · 2022-10-19

**Confidence:** 3
**Correctness:** 3
**Technical Novelty And Significance:** 4
**Empirical Novelty And Significance:** 3
**Recommendation:** 8

**Clarity, Quality, Novelty And Reproducibility:**

The theoretical contribution in this paper is solid and original, and it should be of interest to researchers in both the universal adversarial perturbation, backdoor domains, and robustness certifications. And empirical evaluations took different perspectives into account and lead to some intriguing findings and implications of the robustness certification against UPs.

**Strength And Weaknesses:**

Strength:

1.	The paper is generally well-written and easy to follow, with a well-designed structure;

2.	The problem of robustness certification to UPs is of great importance, and detailed implications to model/defense comparison and backdoor detection are discussed with empirical results;

3.	The proposed method with MILP and linear relaxation accounting for establishing the information sharing between samples is intuitive but compelling; the tightness of the certification results is shown with a comprehensive study with existing certification techniques and actual attacks;

4.	To the best of my knowledge, UP’s generalization analysis in this paper is the first of its kind effort, which also leads to an intriguing analysis of the error of certification results using the proposed robustness certification method.

Weakness:

1.	Some settings of the experiment are unclear. For example, I think the settings for the Implication to backdoor defenses (i.e., the backdoored class identification) can be further improved with more details on the threat model and how to use the proposed tool for detection regarding existing backdoor attacks.

2.	I can understand the difficulties of the robustness certification to large neural networks or large datasets, especially the problem of certifying against UPs requiring computing w.r.t. multiple samples (it seems it is required to compute a much larger computational graph when the network size and input size grow). I highly recommend the authors add a section discussing these limitations.


**Summary Of The Paper:**

This paper looks at a critical but rarely studied problem: the certification of robustness to Universal Perturbations (UPs). In particular, UPs (e.g., universal adversarial noise, backdoor, or neural trojan attacks) have become alarming lines of threats that make it hard to use Machine Learning as a service safely and reliably.

The authors formulated and solved the robustness certification for UPs utilizing a method based on linear relaxation and Mixed Integer Linear Programming. An intriguing discussion on the difference between existing sample-wise certification and the certification against UPs was provided. The authors also theoretically analyzed the relationship between the certification result based on observed samples and the actual robustness of neural networks to UPs w.r.t. the underlying distribution. Multi-perspective evaluations of the tightness (compared with existing certification), soundness of the bound (real UAP and backdoor attacks), and potential implications (to existing attacks and defenses) are offered.

**Summary Of The Review:**

In summary, the authors study a promising robustness certification method against UPs, along with theoretical analysis and a thorough discussion of the implications to model structure, UAP attacks, and backdoor attacks. Some minor issues with the clarity of the settings and the discussion of the limitations should be further improved.

---

> ### Author Response · Authors · 2022-11-09
> **Response to Reviewer GEmp**
>
> Respected Reviewer GEmp,
>
> We would like to thank you for your insightful comments and recognition of this work as the **first-of-its-kind, well-written, fundamental work with compelling method design and important theoretical contributions**.
>
> - In response to your suggestions, we have updated [Appendix B](https://openreview.net/pdf?id=7GEvPKxjtt) to include the experimental settings and threat model for the backdoor identification experiment.
> - In [Appendix D.2](https://openreview.net/pdf?id=7GEvPKxjtt), we have added a discussion on the limitations of generalizing robustness certification to larger scales (datasets or models).
>
> We once again value your feedback and recognition.

---

### Official Review · Reviewer_e7K7 · 2022-10-22

**Confidence:** 5
**Correctness:** 3
**Technical Novelty And Significance:** 4
**Empirical Novelty And Significance:** 4
**Recommendation:** 8

**Clarity, Quality, Novelty And Reproducibility:**

The structure of the paper is clear and easy to read. This work looks into a vital problem but not addressed before in the literature. This study makes some novel and exciting technical contributions to the understanding of UP's generalizability, leading to a solid evaluation of the error bound of the proposed UP-robustness certification. Section 5.3's additional empirical discussion on instantiating UP-robustness certifications to compare model structures, UAP defenses, and to use them as a practical backdoor defense is inspiring and engaging.

**Strength And Weaknesses:**

Strength:

1. This paper focused on a critical problem, certifying the robustness of neural networks to UPs;
2. The paper is well-written. I enjoyed reading this work;
3. A clear structure for recognizing the road map of existing work in UAPs, backdoors, and robustness certification, which facilitates readers to position this work in an interesting intersection between UAP attacks and backdoor attacks;
4. Interesting and solid technical efforts. Given the lack of theoretical analysis in existing UAP and backdoor attacks, the proposed theoretical analyzing framework in this paper gives an interesting first attempt toward a better understanding of neural networks' robustness against these two types of threats;
5. Interesting experimental design. In particular, the author thoroughly evaluated the proposed UP certification and compared it with existing sample-specific certifications, and validated the results with three actual UAP attacks and two types of backdoor attacks. They also devised the theoretical efforts in practice and studied three implications of UP-robustness certification;

Weaknesses:

1. I find the intersection/similarity/difference between UAP attacks and backdoor triggers extended in this work quite interesting. Apart from the current related work section, I highly recommend the authors add a discussion to talk about the similarity/difference between these two lines of threats. Especially some of the recent work in the backdoor literature utilize this interesting intersection for stronger attack efficacy and better stealthiness [1,2] or effective defenses [3].
2. I find most of the experimental settings well-detailed, except for the backdoor target-class identification experiment (Section 5.3). Please add more details to the threat model and the availability of the defender's knowledge, e.g., data accessibility.
3. There is a limitation when using the proposed method to compare different UAP defenses regarding large norms (Table 7, Appendix C.1), i.e., all the results except the result on the IBP-trained model are trivial. I encourage the authors to add discussions on this limitation.

References

[1] Zhang, Quan, et al. "AdvDoor: adversarial backdoor attack of deep learning system." Proceedings of the 30th ACM SIGSOFT International Symposium on Software Testing and Analysis. 2021.

[2] Zeng, Yi, et al. "NARCISSUS: A Practical Clean-Label Backdoor Attack with Limited Information." arXiv preprint arXiv:2204.05255 (2022).

[3] Kolouri, Soheil, et al. "Universal litmus patterns: Revealing backdoor attacks in cnns." Proceedings of the IEEE/CVF Conference on Computer Vision and Pattern Recognition. 2020.


**Summary Of The Paper:**

This paper focused on exploring and extending linear-relaxation-based sample-wise robustness certification to evaluate the robustness of trained neural networks against Universal Perturbations (UPs). The authors pointed out an interesting intersection between universal adversarial perturbations (UAPs) and backdoor triggers. They showcased how the robustness certification to UPs of neural networks can impact these two lines of effort. The theoretical contribution evaluates the generalizability of the effects of the UPs computed using observed samples to unseen data populations. Detailed empirical evaluation regarding multiple perspectives of the proposed method is presented on two standard computer vision benchmarks (MNIST and CIFAR-10) regarding five different model structures (Conv-small, Conv-4-layer, Conv-big, ResNet-2B, and ResNet-4B)

**Summary Of The Review:**

This work resides at the interesting crossroads of UAP and backdoor attacks. The authors presented a novel certification methodology for UP-robustness together with a theoretical study of the result's generalizability. With empirical findings, the implications of this work are discussed in detail. This work has a number of intriguing technical/empirical findings that could lead to future work on backdoor defenses, UAP defenses, and more advanced robustness certification for these threats.

---

> ### Author Response · Authors · 2022-11-09
> **Response to Reviewer e7K7**
>
> Respected Reviewer e7K7,
>
> We would like to express our appreciation for your insightful comments and recognition of this work as an **important, well-written, technically innovated, and empirically thoroughly evaluated work**.
>
> - In response to your comments, we have added an additional discussion to [Appendix D.1](https://openreview.net/pdf?id=7GEvPKxjtt) regarding the interesting intersection of UAP and backdoor attacks.
> - The experimental settings for the backdoor identification experiment are now provided in [Appendix B](https://openreview.net/pdf?id=7GEvPKxjtt).
> - In [Appendix D.2](https://openreview.net/pdf?id=7GEvPKxjtt) is an additional discussion regarding the limitations of robustness certification against large normed perturbations.
>
> Again, we appreciate your feedback and recognition.

---

> > ### Comment · Reviewer_e7K7 · 2022-11-20
> > **Thanks for the response**
> >
> > The reviewer would like to thank the authors for the informative response. I'm now satisfied with the response and the latest version.

---

### Official Review · Reviewer_wZWn · 2022-10-25

**Confidence:** 4
**Correctness:** 2
**Technical Novelty And Significance:** 2
**Empirical Novelty And Significance:** 2
**Recommendation:** 5

**Clarity, Quality, Novelty And Reproducibility:**

· The contributions and the novelty are limited. This paper proposes a
certification defense against universal adversarial examples by modifying
existing certification methods for adversarial examples, i.e., "auto LiRPA".
The only difference is that the proposed method adds a constraint to ensure
the perturbation is global-wise. The technical challenges of the modification
(adding the global-wise constraint) are unclear, and it does not seem very
challenging. Thus, the novelty of the proposed method might be limited.

· This paper claims it can be applied to backdoor attacks. The certification
method uses $L_\infty$ norm to bound the perturbations. However, most backdoor
attacks are not constrained by $L_\infty$ bound. For example, the patch
trigger in BadNets can have a large $L_\infty$ norm. Thus, it is unclear if
the proposed method is able to certify the robustness against backdoor
attacks. I think only a limited number of backdoors can be certified.

· Only empirical defenses for backdoor attacks are discussed. The discussion
about existing certification methods against backdoor attacks (Wang et al.
[1]) is missing. This paper also lacks empirical comparisons between the
proposed method and the existing certification method against backdoor
attacks.

· Generalization to different datasets and models is unclear. Existing
certification work Xu et al. [2] demonstrated it can generalize to different
datasets, including ImageNet and various models (e.g., DenseNet, Transformer
and LSTM). However, in this paper, all experiments are conducted on two
small-scaled datasets (MNIST and CIFAR-10), and it only uses self-defined small
CNNs and ResNet.

· While this paper claims the proposed method is general for different UPs, it
lacks the discussion and the experiments on an important type of attack, i.e.,
patch-based adversarial examples [3]. Thus, it is unclear if the proposed
method is general.

[1] Wang et al., On Certifying Robustness against Backdoor Attacks via Randomized Smoothing. AML@CVPR 2020.
[2] Xu et al., Automatic Perturbation Analysis for Scalable Certified
Robustness and Beyond. NeurIPS 2020.
[3] Tom et al., Adversarial Patch. arXiv 2017.

**Strength And Weaknesses:**

Strengths

· Interesting topic and good motivation.
· The proposed method has better results than sample-Wise certification.
· The writing is good.

Weaknesses

· The novelty might be limited.
· It's unclear if the proposed method can be applied to certify the robustness
against backdoor attacks.
· Comparisons with some related works are missing.
· Generalization of the proposed method is unclear.
· Experiments on adversarial patch attacks are missing.

**Summary Of The Paper:**

This paper proposes a certification defense against universal adversarial
examples and backdoor attacks. The proposed method is based on the combination
of linear relaxation-based perturbation analysis and Mixed Integer Linear
Programming. It also provides a theoretical framework for analyzing the
generalizability of the certification results. Experiments demonstrate that it
is better than existing sample-wise certification methods when defending
against UAPs.

**Summary Of The Review:**

My main concern is the novelty might be limited, and the claim that the
proposed method can be applied to certify the robustness against backdoor
attacks might not be well supported.

---

> ### Author Response · Authors · 2022-11-09
> **Response to Reviewer wZWn**
>
> Respected reviewer wZWn,
>
> We appreciate your feedback and **recognition of the motivation and the quality** of this work.
>
> Below, we provide further details regarding your concerns in a point-to-point manner. We hope the provided information can address your concerns, and we welcome additional discussion if anything remains unclear 😊.

---

> > ### Author Response · Authors · 2022-11-09
> > **Q5: [Experiments on adversarial patch attacks]**
> >
> > **Q5: [Experiments on adversarial patch attacks are missing.]**
> >
> > ***
> >
> > **A:** We focus on providing robustness certification results w.r.t. a given **$l_{\infty}$-norm constraint**. Indeed, **all** certification procedures (including those for sample-wise perturbations) **require some constraints** on potential perturbations and **do not support certification towards unbounded perturbations** w.r.t. to the defined constraint. Tom et al. is not bounded w.r.t. the $l_{\infty}$ constraint, and the results will be non-informative (similar to the results evaluated with the BadNets backdoor in our paper). We have included this reference and discussed the scope of this paper in [Appendix D.2](https://openreview.net/pdf?id=7GEvPKxjtt).

---

> > ### Author Response · Authors · 2022-11-09
> > **Q4: [Generalization of the proposed method]**
> >
> > **Q4: [Generalization of the proposed method is unclear.]**
> >
> > ***
> >
> > **A:**
> > ① Since we adapted auto_LiRPA by (Xu et al., 2020) to obtain the linear bounds in [Section 3.2](https://openreview.net/pdf?id=7GEvPKxjtt), we can similarly obtain the linear bounds for any network architecture as supported by (Xu et al., 2020), and our MILP formulation is independent of the network architecture after the linear bounds are obtained. Therefore, our method in principle supports general network architectures.
> >
> > ② However, supporting larger datasets and larger models remains challenging for certifying pre-trained models, as we need to bound the whole network including a large number of intermediate neurons. As stated in [Section 5.1](https://openreview.net/pdf?id=7GEvPKxjtt), the model structures we selected follow existing state-of-the-art robustness certification work such as [[6](https://proceedings.neurips.cc/paper/2021/hash/fac7fead96dafceaf80c1daffeae82a4-Abstract.html),[7](https://openreview.net/forum?id=l_amHf1oaK)], which also focused on feedforward CNN or ResNet models. The models are not defined by us.
> >
> > ③ (Xu et al., 2020) demonstrated larger models and datasets, but they conducted certified training with relatively cheap Interval Bound Propagation (IBP) to bound intermediate bounds, as the training could optimize the network and tighten the bounds by IBP, which is not applicable to certifying pre-trained models.  Our focus is on solving fundamental challenges in certification against universal perturbations, which is also different from (Xu et al., 2020) which focused on the generalization to various architectures. Supporting various architectures can be viewed as a potential application of our work and is not our focus.
> >
> > [6] Wang, S., Zhang, H., Xu, K., Lin, X., Jana, S., Hsieh, C. J., & Kolter, J. Z. ''Beta-crown: Efficient bound propagation with per-neuron split constraints for neural network robustness verification.'' NeurIPS (2021).
> >
> > [7] Ferrari, C., Mueller, M. N., Jovanović, N., & Vechev, M. ''Complete Verification via Multi-Neuron Relaxation Guided Branch-and-Bound.'' ICLR (2021).

---

> > ### Author Response · Authors · 2022-11-09
> > **Q3: [Comparisons with Wang et al.]**
> >
> > **Q3: [Comparisons with some related works are missing.]** *{Comparison with Wang et al.}*
> >
> > ***
> >
> > **A:** *(Our goals and settings are fundamentally different from those of Wang et al., so we argue that they are not comparable.)*
> >
> > Thanks for pointing us to Wang et al. While they aim to establish robustness guarantees under backdoor attacks, their guarantees are fundamentally different from ours and therefore are **not comparable** to our work. Specifically, the technique in Wang et al. quantifies the test example’s prediction changes when a certain portion of **training samples** is perturbed, but our algorithm quantifies the test accuracy changes when **test samples** are perturbed. In addition to the difference in certification goals, our work also fundamentally differs from theirs in several other aspects:
> >
> > *
> > **[**
> > *Perturbation sharing*
> > **]**
> > The technique in Wang et al. does not consider the fact that a backdoor trigger is a shared perturbation across different samples and adds independent perturbation to different training samples, whereas our method is the first attempt to incorporate the shared perturbation into the design of certification process.
> >
> > *
> > **[**
> > *Agnostic to learning algorithm*
> > **]**
> > The technique in Wang et al. requires modifying the learning algorithm by adding random noise to training samples and feeding noised samples into the training algorithm and can only certify the robustness of this modified learning algorithm. However, our certification method is agnostic to the underlying learning algorithm. It reasons about the robustness level of a trained model (as explained in Section 2, discussing the difference with smoothness-based certification methods, we now have also included Wang et al.).
> >
> > *
> > **[**
> > *Efficiency*
> > **]**
> > In addition to adding random noise to training data, the technique in Wang et al. also requires retraining the model on noisy samples multiple times, which incurs large computation costs. In fact, their method has only been applied to a binary MNIST classification task (''1'', ''7'' classification with all the pixel values being simplified to 0s or 1s) with toy models that only have two layers. By contrast, our method does not require retraining. The most expensive part of our certification process is solving MILP; with the off-the-shelf solvers, our method can easily scale to standard image data benchmarks, like CIFAR-10.

---

> > ### Author Response · Authors · 2022-11-09
> > **Q2: [Can the proposed method being applied to (un-bounded) backdoor attacks?]**
> >
> > **Q2: [It's unclear if the proposed method can be applied to certify the robustness against (un-bounded) backdoor attacks.]**
> >
> > ***
> >
> > **A:** We agree with your comments on the limitation of applying robustness certification to un-bounded backdoor attacks. However, **this limitation is not unique to the certification of UPs**. In fact, **any certification procedure** (including those for sample-wise perturbations) **requires some constraints on potential perturbations** and **does not apply to unbounded perturbations**. Especially, the results on un-bounded attacks might be the same for clean models and backdoored models and thus non-informative (as discussed in [Section 5.2](https://openreview.net/pdf?id=7GEvPKxjtt) with an unbounded BadNets trigger).
> >
> > However, as the first robustness certification works toward UPs, we emphasize that **robustness certification to UPs is essential to backdoor attacks.** In particular, we have shown a successful identification of a potential backdoor with $l_{\infty}$-bounded backdoor in Figure [4](https://openreview.net/pdf?id=7GEvPKxjtt), and we also elaborate a specific downstream usage of reliable target class identification using the certification method with three backdoor attacks in [Section 5.3](https://openreview.net/pdf?id=7GEvPKxjtt) and [Appendix C.2](https://openreview.net/pdf?id=7GEvPKxjtt). Additionally, we would like to point out there is an ongoing trend of making backdoor triggers more and more stealthy with additional optimization constraints, especially $l_{\infty}$ constraints [[3](https://arxiv.org/abs/1912.02771),[4](https://arxiv.org/abs/2106.08970),[5](https://arxiv.org/abs/2204.05255)]. Thus, robustness certification towards UPs is of great importance to **understanding backdoor attacks** and **provides a pathway to identify models that are potentially poisoned with backdoors.**
> >
> > [3] Turner, Alexander, Dimitris Tsipras, and Aleksander Madry. "Label-consistent backdoor attacks." arXiv preprint arXiv:1912.02771 (2019).
> >
> > [4] Souri, Hossein, Micah Goldblum, Liam Fowl, Rama Chellappa, and Tom Goldstein. "Sleeper agent: Scalable hidden trigger backdoors for neural networks trained from scratch." NeurIPS (2022).
> >
> > [5] Zeng, Yi, Minzhou Pan, Hoang Anh Just, Lingjuan Lyu, Meikang Qiu, and Ruoxi Jia. "NARCISSUS: A Practical Clean-Label Backdoor Attack with Limited Information." arXiv preprint arXiv:2204.05255 (2022).

---

> > > ### Comment · Reviewer_wZWn · 2022-11-17
> > > **Clarification about my comments**
> > >
> > > I want to clarify that I did not ask to apply the method to unbounded backdoor attacks. Most backdoor attacks are bounded, but not in $L_\infty$ bound. Thus, the claim that this method can be used to certify backdoor attacks is not well-supported.

---

> > > > ### Author Response · Authors · 2022-11-17
> > > > **Certification Results to $l_{\infty}$-Bounded Backdoors are Still Important**
> > > >
> > > > Huge thanks for the clarification. We have updated our manuscript to ensure our claim only centered on certifying $l_{\infty}$-norm-bounded backdoor attacks (not all backdoors). However, as we have stated, the **certification results towards $l_{\infty}$-bouned backdoors are still of great importance** to the backdoor community as:
> > > >
> > > > *
> > > > Existing works have found that $l_{\infty}$ unbounded triggers (e.g., BadNets, $l_{0}$-invisble attack, $l_{2}$-invisble attack [a]) are relatively easier to be detected with noticeable artifacts [b] or spectrum traces [c] than those bounded ones (e.g., classic example as the Blended [d], which have already included and evaluated in Section 5.2).
> > > >
> > > > *
> > > > The ongoing trend in the backdoor community is to **develop more stealthy backdoor attacks**. Among these lines of novel backdoor attacks emphasizing "stealthiness," **a critical line of work is the clean-label backdoor attacks** whose poisoning process does not include label-flipping [c, d, e, f]. In particular, **all four representative works mentioned here are $l_{\infty}$-norm-bounded backdoor attacks.**
> > > >
> > > > Additionally, we have included a short discussion on the limitation of applying our method on certifying unconstrained or large $l_{\infty}$-norm backdoor attack in Appendix D.2.
> > > >
> > > > ***
> > > > [a] Li, Shaofeng, et al. "Invisible backdoor attacks on deep neural networks via steganography and regularization." IEEE TDSC 18.5 (2020): 2088-2105.
> > > >
> > > > [b] Zeng, Yi, et al. "Rethinking the backdoor attacks' triggers: A frequency perspective." ICCV'21.
> > > >
> > > > [c] Xiang, Zhen, David J. Miller, and George Kesidis. "Post-training detection of backdoor attacks for two-class and multi-attack scenarios." ICLR'22.
> > > >
> > > > [d] Chen, Xinyun, et al. "Targeted backdoor attacks on deep learning systems using data poisoning." arXiv preprint arXiv:1712.05526 (2017).
> > > >
> > > > [c] Saha, Aniruddha, Akshayvarun Subramanya, and Hamed Pirsiavash. "Hidden trigger backdoor attacks." AAAI'20.
> > > >
> > > > [d] Turner, Alexander, Dimitris Tsipras, and Aleksander Madry. "Label-consistent backdoor attacks." arXiv preprint arXiv:1912.02771 (2019).
> > > >
> > > > [e] Souri, Hossein, et al. "Sleeper agent: Scalable hidden trigger backdoors for neural networks trained from scratch." NeurIPS'22.
> > > >
> > > > [f] Zeng, Yi, et al. "NARCISSUS: A Practical Clean-Label Backdoor Attack with Limited Information." arXiv preprint arXiv:2204.05255 (2022).

---

> > ### Author Response · Authors · 2022-11-09
> > **Q1: [Novelty]**
> >
> > **Q1: [Novelty]**
> > ***
> > **A:**
> > ① *Formulating shared information of the universal perturbation is challenging;*
> >
> > $\ \ \ $ ② *The bound propagation and MILP used in this paper have a novel perspective and functionality than what has previously been published;*
> >
> > $\ \ \ $ ③ *We would also like to emphasize the novelty of this work on theoretically analyzing the generalizability of the UP attacks' efficacy and certification results.*

---

> > > ### Author Response · Authors · 2022-11-09
> > > **③ Novel Theoretical Contributions**
> > >
> > > ③ In addition to the innovation above in the design of our verification method, we present the first theoretical analysis framework of the generalizability of certification against UPs for a given neural network. Note that existing verification methods are typically sample-wise (e.g., certifying whether or not the network is robust to perturbations added on a given sample), and thus they do not require reasoning about the generalizability of the certification outcome to the underlying data distribution. In this work, we formalized a series of novel concepts (such as attack rate, and attack probability) to reason about generalization and established the first generalization bound for UP certification. We believe these efforts will be helpful to catalyst future advances in UP certification.

---

> > > ### Author Response · Authors · 2022-11-09
> > > **② Novel Methodology Design**
> > >
> > > ② The use (combination) of bound propagation and MILP in our work differs from any existing certification works.
> > >
> > > *
> > > **[**
> > > *Bound propagation with perturbation sharing.*
> > > **]**
> > > As explained in Section 3.2, we use bound propagation to obtain linear bounds w.r.t. a shared perturbation $\delta$, instead of the input of each sample independently as considered in the previous bound propagation works, such that the perturbation can be shared when we concretize the linear bounds by MILP.
> > >
> > > *
> > > **[**
> > > *Our use of MILP is significantly different from the previous certification works.*
> > > **]**
> > > MILP in the previous certification works involves integer variables to model the state of ReLU activations [[1](https://arxiv.org/abs/1711.07356)]. We have a significant intent of using MILP and a significantly different formulation. Our integer variables model whether each example is correctly predicted and whether a class is the hardest among all the incorrect classes (the second most likely class) for each sample, respectively, under a universal perturbation.
> > >
> > > [1] Tjeng, Vincent, Kai Y. Xiao, and Russ Tedrake. "Evaluating Robustness of Neural Networks with Mixed Integer Programming." In International Conference on Learning Representations. 2018.

---

> > > ### Author Response · Authors · 2022-11-09
> > > **① Novel Formulation Resolving Challenges of ''Infomation Share''**
> > >
> > > ① It is well said that a key difference between this work’s proposed certification method to existing certification methods is the added constraint ensuring the perturbation to be global-wise. However, we argue that formulation of such an ``information share'' is indeed challenging, and no existing work has successfully done so. Throughout our design, we have encountered numerous new technical challenges that have required us to propose novel solutions while keeping the formulation tractable:
> > >
> > >
> > > ***
> > > At a high level, our goal is to calculate worst-case accuracy under a shared perturbation $\delta$ across all samples. To achieve this, we introduced binary variables $s^{(i)}\_j$ to indicate whether $j$ is the second most likely class index for a sample $z_i$. Also, we introduced $q^{(i)}$, which indicates whether one sample is correctly classified. $s^{(i)}\_j$ and $q^{(i)}$ can be specified through if-else statements:
> > > \begin{aligned}
> > > \min &\sum\_i q^{(i)} \quad\quad\quad\quad\quad\quad\quad\quad\quad\quad\quad\quad\quad\quad\quad\quad\quad\quad\quad\quad\quad\quad\quad\quad\quad\quad\quad\quad\quad\quad\quad\quad\quad\quad\quad\quad(1)
> > > \\\\
> > > \text{s.t. } &q^{(i)} = 1, \text{if } \sum\_{j\neq y_i} \mathbf{m}\_{y_i,j}(
> > > \mathbf{x}_i+\delta
> > > ) s_j^{(i)} > 0
> > > \text{ (i.e., sample $\mathbf{x}$ returns the correct label w.r.t. $\delta$)}; \quad\qquad\\,(2)
> > > \\\\
> > > &q^{(i)} = 0, \text{else if } \sum\_{j\neq y_i} \mathbf{m}\_{y_i,j}(
> > > \mathbf{x}_i+\delta
> > > ) s_j^{(i)} \leq 0
> > > \text{ (i.e., sample $\mathbf{x}$'s output is perturbed w.r.t. $\delta$)}; \\ \qquad\ \\,(3)
> > > \\\\
> > > &s^{(i)}_j = 1, \text{if $j$ is the index of the second largest logit}; \quad\quad\quad\quad\quad\quad\quad\quad\quad\quad\quad\quad\quad\quad\quad\quad\quad\\: \\,(4)
> > > \\\\
> > > &s^{(i)}_j = 0, \text{else if $j$ is not the index of the second largest logit};
> > > \quad\quad\quad\quad\quad\quad\quad\quad\quad\quad\quad\quad\quad\\ \ \ \\:(5)
> > > \\\\
> > > &||\delta||\_{\infty} \leq \epsilon \quad\quad\quad\quad\quad\quad\quad\quad\quad\quad\quad\quad\quad\quad\quad\quad\quad\quad\quad\quad\quad\quad\quad\quad\quad\quad\quad\quad\quad\quad\quad\quad\quad\quad\quad\\,(6)
> > > \end{aligned}
> > > However, there are several technical challenges associated with this natural formulation:
> > >
> > > *
> > > **[**
> > > *Efficiency*
> > > **]**
> > > $\mathbf{m}\_{y_i,j}$ is the logit margin between class $y_i$ and $j$, and expressing it exactly requires the optimization to encode the whole neural network to be certified. When the network uses ReLU activation units, it is possible to express the network as linear integer constraints directly.
> > > However, it will be extremely computationally prohibitive. Even for very small networks with thousands of neurons, the number of integer variables in their MILP formulation will be proportional to the number of neurons. We tackle this limitation by first computing linear bounds for the neural network; as a result, the number of integer variables in our formulation does not depend on the size of the network, which makes it feasible in practice.
> > >
> > > *
> > > **[**
> > > *Handling novel constraints*
> > > **]**
> > > The if-else type constraints in Eqn. (2)&(3) or Eqn. (4)&(5) cannot be solved by any standard optimization solvers. Moreover, the constraints contain strict inequalities are hard to deal with in optimization (when the feasible set is an open set, there exists convergence issues). To make our formulation tractable, we leverage the Big M method technique to convert the if-else statements to linear constraints (See Eqn. 6 and 8 in [our paper](https://openreview.net/pdf?id=7GEvPKxjtt)).

---

> ### Author Response · Authors · 2022-11-16
> **A Friendly Reminder [rebuttal update portal will close in three days]**
>
> Dear Reviewer wZWn,
>
> We like to reiterate our gratitude for your comments and insights. This is a kindly reminder that the update link for the author's response will conclude in three days.
>
> We have offered a detailed answer to your concerns. Please let us know if any of your concerns were not adequately addressed. We would be delighted to incorporate your comments into the rebuttal update.
>
> Best,
>
> Authors

---

> ### Author Response · Authors · 2022-11-17
> **With the hope that our response addresses your concerns**
>
> Dear Reviewer wZWn,
>
> We appreciate you taking the time to review our work. We would appreciate your feedback on whether our responses addressed your concern (especially on novelty). As the conclusion of the rebuttal phase approaches, we anticipate hearing from you and stand ready to provide any additional clarification you may require.
>
> Thank you beforehand,
>
> Authors of Paper 1550

---

> ### Author Response · Authors · 2022-11-19
> **Discussion period ending–we anticipate your feedback!**
>
> Dear Reviewer wZWn,
>
> As the discussion period is closing, we sincerely look forward to more of your feedback. The authors sincerely appreciate your valuable time and efforts spent reviewing this paper and helping us improve it.
>
> It would be very much appreciated if you could once again help review our responses and let us know if these address or partially address your concerns and if our explanations are heading in the right direction.
>
> Please also let us know if there are further questions or comments about this paper. We strive to improve the paper consistently, and it is our pleasure to have your feedback!
>
> Kind Regards,
>
> Authors of Paper1550

---

> ### Author Response · Authors · 2022-11-26
> **We anticipate your feedback!**
>
> Dear Reviewer wZWn,
>
>
> The conclusion of the discussion period is closing, and we eagerly await your response. The authors greatly appreciate your time and effort in reviewing this paper and helping us improve it.
>
> We have provided detailed responses to every one of your concerns. Please help us to review our responses once again and kindly let us know whether they fully or partially address your concerns and if our explanations are in the right direction.
>
>
>
> Kind Regards,
>
> Authors of Paper1550

---

> > ### Comment · Reviewer_wZWn · 2022-11-26
> > **Feedback**
> >
> > Thanks for the comments. I think the paper has great potential. Some more effort into its novelty and significance will make it publishable.

---

> > > ### Author Response · Authors · 2022-11-28
> > > **Sincere thanks for your review and valuable feedback.**
> > >
> > > We'd like to thank the reviewer's efforts in providing valuable feedback.
> > >
> > > We want to take this opportunity to clarify that **while the proposed certification method is built upon linear perturbation bounds proposed in the prior sample-wise certification literature, our optimization formulation for certifying against UPs is anything but a trivial extension.** Below, we'd like to highlight some **key differences from sample-wise certification and our unique novelties**, which entail new technical challenges, new solution methods, and new results:
> > >
> > > *
> > > In sample-wise certification, the goal is to inspect whether the logit of the ground-truth label for a given test point is larger than the rest of the classes. By contrast, in UP certification, the goal is to certify the aggregate accuracy on a batch of samples, and therefore, we need to track whether the most likely class's logit for each sample in a batch exceeds the second most likely class under a shared perturbation. In other words, there is a unique challenge here of **tracking the second most likely class, which does not exist for sample-wise certification.** **Identifying the second most likely class introduces if-else type constraints (e.g., if the logit ranks the second, then it corresponds to the second most likely class), and importantly, off-the-shelf optimization solvers cannot handle such constraints.** To tackle this challenge, we encode this constraint using integer variables and leverage techniques (such as the Big M method) to ensure the overall formulation can be solved with off-the-shelf solvers.
> > > *
> > > Unlike sample-wise perturbation, certification for UPs requires **understanding how the certification result obtained from a batch of samples generalizes to the underlying data distribution.** This is a **unique challenge** of UPs, and we provide the **first theoretical framework** to enable the analysis of generalization, and the bound is remarkably **tight**, as evidenced in Figure 2. This theoretical result is particularly significant in analyzing UAP and backdoor attacks to a trained model, and it can be easily adopted in any following work.
> > > *
> > > Novelty and significance reside in further steps in empirical exploration: As the first work on certifying the robustness of UPs, we also investigated three downstream implication explorations with the proposed certification method: we showcase how UP-robustness certification **1)** can help to compare the robustness of different model structure; **2)** can help to compare different existing UAP defenses; **3)** can be twisted to provide reliable backdoor detection and identification of the target class. Especially the backdoor identification task using the proposed certification framework is **unlike anything presented** in the literature.
> > >
> > >
> > > As we have already stated and recognized by multiple reviewers, this work is the first focused effort to certify a trained model's robustness against norm-bounded UPs. **This work is significant as it sets the foundation for UP certification by providing an efficient certification algorithm as well as novel concepts and theoretical frameworks for reasoning about generalization.** In particular, our framework can be extended to improve certifiable robustness against UP by using our certification as the training goal.

---

### Official Review · Reviewer_Kpjw · 2022-10-25

**Confidence:** 3
**Correctness:** 3
**Technical Novelty And Significance:** 2
**Empirical Novelty And Significance:** 2
**Recommendation:** 5

**Clarity, Quality, Novelty And Reproducibility:**

The paper is well written and presented. It is highly incremental to previous
work as existing methods on bound propagation and MILP formulations are almost
entirely used as appeared in previous works.  I think that the novelty can be
improved by exploiting the restricted nature of the problem studied (which
couples the perturbation radius to all inputs) to improve the scalability of
verification.

**Strength And Weaknesses:**

+ The paper is the first to the best of my knowledge to discuss the formal
verification of universal perturbations.

+ The paper includes an analysis of the generalisation of robustness results
from samples to the entire data distribution.

- The verification method is a straightforward adaptation of existing methods.

- Unsound comparison with bound propagation-based methods: whereas the
  certified robust rate for the bound propagation methods is taken as the rate
  of images for which a model is robust, said rate for the current method is
  being minimised in an MILP formulation.


**Summary Of The Paper:**

The paper studies the verification of neural networks against universal
adversarial perturbations, i.e., perturbations that are universally applied on
a set of images. It solves the verification problem by adapting bound
propagation- and MILP-based verification methods to  universal perturbations.

**Summary Of The Review:**

Whilst the paper is the first to study the verification problem for neural
networks and  universal adversarial perturbations, the resulting method is
highly incremental to previous work and the soundness of the comparisons with
related work is not convincing.

---

> ### Author Response · Authors · 2022-11-09
> **Response to Reviewer Kpjw**
>
> Respected reviewer Kpjw,
>
> We value your feedback and your recognition of this work as the **first work to study the important problem of verifying the robustness of a given neural network to universal perturbations (UPs)**. We respectfully disagree with your comments regarding this work's novelty and the comparisons' validity.
>
> We will expand on our discussions below, and we hope that you can find your concerns adequately addressed. If there is anything that remains unclear, we would appreciate a follow-up discussion 😊.

---

> > ### Author Response · Authors · 2022-11-09
> > **Q2: [Validity of the comparison]**
> >
> > **Q2: [Validity of the comparison]**  *{Unsound comparison … bound propagation methods is taken as the rate of images for which a model is robust …}*
> >
> > ***
> >
> > **A:**  *(The comparison is sound and fair as the problem to solve is the same. But the sample-wise bound propagation methods provide a lower bound of our formulation.)*
> >
> > We argue that the comparison is sound indeed, as both the comparison group and our method attempt to resolve the same problem of obtaining the certified robust rate of a given batch of data towards universal perturbations. Specifically, note that
> > $$
> > \min_{{||\delta||}_p \leq \epsilon} \frac{1}{n} \sum\_{i=1}^n \mathbb{1}
> > \bigg(
> > \min\_{j \neq y_i} \\{
> > \mathbf{m}\_{y_i, j} (\mathbf{x}_i+\delta) > 0
> > \\}
> > \bigg)
> > \geq
> > \frac{1}{n}
> > \sum\_{i=1}^n
> > \min\_{||\delta\_i||_p \leq \epsilon} \mathbb{1}
> > \bigg(
> > \min\_{j \neq y\_i}
> > \\{
> > \mathbf{m}\_{y_i,j}(\mathbf{x}_i+\delta_i)
> > \\}
> > \>0
> > \bigg).
> > $$
> > Hence, sample-wise certified accuracy can be utilized to obtain a lower bound of the accuracy under UP. We believe this is a valid and important baseline to compare with given the lack of other techniques dedicated to UP certification.

---

> > ### Author Response · Authors · 2022-11-09
> > **Q1: [Novelty]**
> >
> > **Q1: [Novelty]**  *{... straightforward adaptation… both bound propagation and MILP have been used as appeared in previous work.}*
> >
> > ***
> >
> > **A:** ① *Our formulation is unique and novel;*
> >
> > $\ \ \ $ ② *The bound propagation and MILP used in this paper have a distinct emphasis and intent than what have previously been published;*
> >
> > $\ \ \ $ ③ *We would also like to emphasize the novelty of this work on theoretically analyzing the generalizability of the UP attacks' efficacy and certification results.*

---

> > > ### Author Response · Authors · 2022-11-09
> > > **③ Novel Theoretical Contributions**
> > >
> > > ③ In addition to the innovation above in the design of our verification method, we present the first theoretical analysis framework of the generalizability of certification against UPs for a given neural network. Note that existing verification methods are typically sample-wise (e.g., certifying whether or not the network is robust to perturbations added on a given sample), and thus they do not require reasoning about the generalizability of the certification outcome to the underlying data distribution. In this work, we formalized a series of novel concepts (such as attack rate, and attack probability) to reason about generalization and established the first generalization bound for UP certification. We believe these efforts will be helpful to catalyst future advances in UP certification.

---

> > > ### Author Response · Authors · 2022-11-09
> > > **② Our Methodology is Novel**
> > >
> > > ② The use (combination) of bound propagation and MILP in our work differs from any existing certification works.
> > >
> > > *
> > > **[**
> > > *Bound propagation with perturbation sharing.*
> > > **]**
> > > As explained in Section 3.2, we use bound propagation to obtain linear bounds w.r.t. a shared perturbation $\delta$, instead of the input of each sample independently as considered in the previous bound propagation works, such that the perturbation can be shared when we concretize the linear bounds by MILP.
> > >
> > > *
> > > **[**
> > > *Our use of MILP is significantly different from the previous certification works.*
> > > **]**
> > > MILP in the previous certification works involves integer variables to model the state of ReLU activations [[1](https://arxiv.org/abs/1711.07356)]. We have a significant intent of using MILP and a significantly different formulation. Our integer variables model whether each example is correctly predicted and whether a class is the hardest among all the incorrect classes (the second most likely class) for each sample, respectively, under a universal perturbation.
> > >
> > > [1] Tjeng, Vincent, Kai Y. Xiao, and Russ Tedrake. "Evaluating Robustness of Neural Networks with Mixed Integer Programming." In International Conference on Learning Representations. 2018.

---

> > > ### Author Response · Authors · 2022-11-09
> > > **① Our Formulation is Unique and Novel**
> > >
> > > ① The formulation of certifying the robustness of a neural network to UPs is fundamentally different than existing formulations on certifying the robustness of **one sample** given a neural network. Throughout our design, we have encountered numerous new technical challenges that have required us to propose novel solutions while keeping the formulation tractable. We argue that our formulation is anything but a straightforward adaptation of existing sample-wise certifications, as detailed below:
> > >
> > > ***
> > >
> > > At a high level, our goal is to calculate worst-case accuracy under a shared perturbation $\delta$ across all samples. To achieve this, we introduced binary variables $s^{(i)}\_j$ to indicate whether $j$ is the second most likely class index for a sample $z_i$. Also, we introduced $q^{(i)}$, which indicates whether one sample is correctly classified. $s^{(i)}\_j$ and $q^{(i)}$ can be specified through if-else statements:
> > > \begin{aligned}
> > > \min &\sum\_i^n q^{(i)} \quad\quad\quad\quad\quad\quad\quad\quad\quad\quad\quad\quad\quad\quad\quad\quad\quad\quad\quad\quad\quad\quad\quad\quad\quad\quad\quad\quad\quad\quad\quad\quad\quad\quad\quad\quad\quad\quad\ \ \ (1)
> > > \\\\
> > > \text{s.t. } & \forall i\in[n], q^{(i)} = 1, \text{if } \sum\_{j\neq y_i} \mathbf{m}\_{y_i,j}(
> > > \mathbf{x}_i+\delta
> > > ) s_j^{(i)} > 0
> > > \text{ (i.e., sample $\mathbf{x}$ returns the correct label w.r.t. $\delta$)}; \qquad\\,(2)
> > > \\\\
> > > & \forall i\in[n], q^{(i)} = 0, \text{else if } \sum\_{j\neq y_i} \mathbf{m}\_{y_i,j}(
> > > \mathbf{x}_i+\delta
> > > ) s_j^{(i)} \leq 0
> > > \text{ (i.e., sample $\mathbf{x}$'s output is perturbed w.r.t. $\delta$)}; \\ \quad\ \\,(3)
> > > \\\\
> > > &\forall i\in[n], s^{(i)}_j = 1, \text{if $j$ is the index of the second largest logit}; \quad\quad\quad\quad\quad\quad\quad\quad\quad\quad\quad\quad\quad\quad\quad\quad\\: \\,(4)
> > > \\\\
> > > &\forall i\in[n], s^{(i)}_j = 0, \text{else if $j$ is not the index of the second largest logit};
> > > \quad\quad\quad\quad\quad\quad\quad\quad\quad\quad\quad\quad\\ \ \ \\,(5)
> > > \\\\
> > > &||\delta||\_{\infty} \leq \epsilon \quad\quad\quad\quad\quad\quad\quad\quad\quad\quad\quad\quad\quad\quad\quad\quad\quad\quad\quad\quad\quad\quad\quad\quad\quad\quad\quad\quad\quad\quad\quad\quad\quad\quad\quad\quad\quad\ \ \\, \\,(6)
> > > \end{aligned}
> > > However, there are several technical challenges associated with this natural formulation:
> > >
> > > *
> > > **[**
> > > *Efficiency*
> > > **]**
> > > $\mathbf{m}\_{y_i,j}(\cdot)$ is the logit margin between class $y_i$ and $j$, and expressing it exactly requires the optimization to encode the whole neural network to be certified. When the network uses ReLU activation units, it is possible to express the network as linear integer constraints
> > > [*We believe the MILP mentioned by the reviewer refers to those prior works (e.g., [[1](https://arxiv.org/abs/1711.07356)]) that introduce integer variables to encode the state of ReLU unit; note that our integers are introduced for a completely different purpose*].
> > > However, it will be extremely computationally prohibitive. Even for very small networks with thousands of neurons, the number of integer variables in their MILP formulation will be proportional to the number of neurons. We tackle this limitation by first computing linear bounds for the neural network; as a result, the number of integer variables in our formulation does not depend on the size of the network, which makes it feasible in practice.
> > >
> > > *
> > > **[**
> > > *Handling novel constraints*
> > > **]**
> > > The if-else type constraints in Eqn. (2)&(3) or Eqn. (4)&(5) cannot be solved by any standard optimization solvers. Moreover, the constraints contain strict inequalities are hard to deal with in optimization (when the feasible set is an open set, there exists convergence issues). To make our formulation tractable, we leverage the Big M method technique to convert the if-else statements to linear constraints (See Eqn. 6 and 8 in [our paper](https://openreview.net/pdf?id=7GEvPKxjtt)).
> > >
> > > [1] Tjeng, Vincent, Kai Y. Xiao, and Russ Tedrake. "Evaluating Robustness of Neural Networks with Mixed Integer Programming." In International Conference on Learning Representations. 2018.

---

> > ### Comment · Reviewer_Kpjw · 2022-11-23
> > **Thank you for the response**
> >
> > I thank the authors for their response. I appreciate the novel contributions
> > the authors discuss regarding the adaptations of SIP and MILP formulations. It
> > still remains my opinion however that these are entirely standard and therefore
> > the novelty is limited. As for the soundness of the comparison, I of course
> > agree with the authors that certified accuracy on standard perturbations
> > provides a lower bound of certified accuracy on universal perturbations. So the
> > former perturbations can be informally viewed as a restricted from of the
> > latter. Thus it is to be expected that certified accuracy on the former would
> > be larger. The paper however may be giving the message that the improved
> > accuracy is observed not because of the aforementioned reason but because of
> > having developed a more efficient method than the SoA in neural network
> > verification, which is not entirely true. Hence the comment in my review. As I
> > however think that the message can be easily corrected I increased my score to 5
> > with my main concern remaining the limited novelty of the work.

---

> > > ### Author Response · Authors · 2022-11-23
> > > **Sincere thanks for your review efforts and valuable feedback.**
> > >
> > > The authors would like to express their most profound appreciation for the reviewer's time, efforts, and insightful feedback.
> > >
> > > Regarding the novelty concern, we would like to bring to your attention **three additional points** in an effort to address your concerns:
> > >
> > > *
> > > MILP is a standard tool for resolving optimization problems indeed. The difficulty of adopting MILP is the **successful formulation** of the problem into a MILP problem, which is also one of the major contributions of this work. The formulation is **intuitive** and **effective**; the simplicity of the proposed method should not be regarded as a limitation of novelty. In particular, we are the **first practical solution** to this challenging problem.
> > > *
> > > Novelty resides in theoretical contributions: Unlike sample-wise perturbation, certification for universal perturbations (UPs) requires understanding how the certification result obtained from a batch of samples generalizes to the underlying data distribution. This is a **unique challenge** of UPs, and we provide the **first theoretical framework** to enable the analysis of generalization, and the bound is remarkably **tight**, as evidenced in Figure 2.
> > > *
> > > Novelty resides in further steps in empirical exploration: As the first work on certifying the robustness of universal perturbations, we also investigated three downstream implication explorations with the proposed certification method: we showcase how UP-robustness certification **1)** can help to compare the robustness of different model structure; **2)** can help to compare different existing UAP defenses; **3)** can be twisted to provide reliable backdoor detection and identification of the target class. Especially the backdoor identification task using the proposed certification framework is **unlike anything presented** in the literature.
> > >
> > >
> > >
> > > Regarding the validity of the comparison, we appreciate the reviewer's clarifications. We will revise the paper and emphasize the justification for using the sample-based result as the baseline.

---

> ### Author Response · Authors · 2022-11-16
> **A Friendly Reminder [rebuttal update portal will close in 3 days]**
>
> Dear Reviewer Kpjw,
>
> We want to express our appreciation for your comments and insights again. This is a friendly reminder that the author's rebuttal update portal will close in three days.
>
> We have provided a point-to-point response to your concerns. Please kindly let us know if you have any concerns you find not fully addressed. We would be more than happy to include your suggestions in the rebuttal update.
>
> Best,
>
> Authors

---

> ### Author Response · Authors · 2022-11-17
> **With the hope that our response addresses your concerns**
>
> Dear Reviewer Kpjw,
>
> Thank you once more for taking the time to review our work! We'd appreciate it if you could let us know whether our response addressed your concern. As the end of the rebuttal phase approaches, we look forward to hearing from you and remain available for any additional clarification you may require.
>
> Thank you in advance,
>
> Authors of Paper 1550

---

> ### Author Response · Authors · 2022-11-19
> **Discussion period ending–we anticipate your feedback!**
>
> Dear Reviewer Kpjw,
>
> As the discussion period is closing, we sincerely look forward to your feedback. The authors deeply appreciate your valuable time and efforts spent reviewing this paper and helping us improve it.
>
> It would be very much appreciated if you could once again help review our responses and let us know if these address or partially address your concerns and if our explanations are heading in the right direction.
>
> Please also let us know if there are further questions or comments about this paper. We strive to improve the paper consistently, and it is our pleasure to have your feedback!
>
> Kind Regards,
>
> Authors of Paper1550

---

### Author Response · Authors · 2022-11-17
**General Response and Rebuttal Summarization**

Dear AC and Reviewers,

The insightful comments and suggestions of all reviewers are greatly valued. We are delighted that our work has been recognized as

*
**the first** formal certification of a model's robustness against universal perturbations (*Kpjw, GEmp*);
*
**innovative** and **substantial** theoretical contribution to the study of the generalization of UPs (*e7K7, GEmp*);
*
**extensive** and **well-designed** empirical analysis (*e7K7, GEmp*);
*
**well-written** (*Kpjw, wZWn, e7K7, GEmp*);
*
**well-motivated** (*wZWn, e7K7, GEmp*).
***
**Updates:** We have revised the manuscript in response to the comments and suggestions of reviewers, and the revised parts are highlighted in $\\textbf{\textcolor{orange}{orange}}$. Here is a summary of the modification:

1. We have made sure all the claims on providing certification on backdoor attacks are centering at "$l_{\infty}$-norm-bounded" backdoors;

2. Details of settings for adapting the proposed method to identify backdoor are provided in Appendix B;

3. Additional discussion on UAP and backdoor attacks are provided in Appendix D.1;

4. Additional discussions on limitation towards unconstrained or large $l_{\infty}$-norm-bounded attacks and computational cost are provided in Appendix D.2.
***
The manuscript's clarity has been enhanced by the reviewers' tremendous efforts and insightful comments. We have thoughtfully addressed the primary concerns and provided detailed responses to each reviewer. We hope you find the responses to your questions satisfactory. Regarding our responses to the reviews, we would greatly appreciate your feedback.

Best,

Authors of Paper 1550

---

### Decision · Program_Chairs · 2023-01-20

**Decision:**

Accept: poster

**Justification For Why Not Higher Score:**

Some reviewers thought the resulting method is incremental to previous work and the soundness of the comparisons with related work is not convincing.

**Justification For Why Not Lower Score:**

some reviewers thought the work has a number of intriguing technical/empirical findings that could lead to future work on backdoor defenses, UAP defenses, and more advanced robustness certification for these threats.

**Metareview: Summary, Strengths And Weaknesses:**

This paper focused on exploring and extending linear-relaxation-based sample-wise robustness certification to evaluate the robustness of trained neural networks against Universal Perturbations (UPs). The authors pointed out a connection between universal adversarial perturbations (UAPs) and backdoor triggers. Reviewers had some mixed thoughts on the work: some reviewers thought the work has a number of intriguing technical/empirical findings that could lead to future work on backdoor defenses, UAP defenses, and more advanced robustness certification for these threats. Other reviewers thought the resulting method is highly incremental to previous work and the soundness of the comparisons with related work is not convincing and the novelty might be limited. Authors did a good job during the rebuttal addressing some of these concerns.  Given all, I think the paper is slightly above the bar!

**Note From Pc:**

if the above contains the word "oral" or "spotlight" please see: "oral" presentation means -> notable-top-5% and "spotlight" means -> notable-top-25%. As stated in our emails, we are disassociating presentation type from AC recommendations